**Article** https://doi.org/10.1038/s41467-023-40148-6

# Cellular state landscape and herpes simplex virus type 1 infection progression are connected

**Maija K. Pietilä** [1] ✉, **Jana J. Bachmann** [1], **Janne Ravantti** [2], **Lucas Pelkmans**[3] & **Cornel Fraefel** [1] ✉

Prediction, prevention and treatment of virus infections require understanding of cell-to-cell variability that leads to heterogenous disease outcomes, but the source of this heterogeneity has yet to be clarified. To study the multimodal response of single human cells to herpes simplex virus type 1 (HSV-1) infection, we mapped high-dimensional viral and cellular state spaces throughout the infection using multiplexed imaging and quantitative single-cell measurements of viral and cellular mRNAs and proteins. Here we show that the high-dimensional cellular state scape can predict heterogenous infections, and cells move through the cellular state landscape according to infection progression. Spatial information reveals that infection changes the cellular state of both infected cells and of their neighbors. The multiplexed imaging of HSV-1-induced cellular modifications links infection progression to changes in signaling responses, transcriptional activity, and processing bodies. Our data show that multiplexed quantification of responses at the single-cell level, across thousands of cells helps predict infections and identify new targets for antivirals.

Single-cell heterogeneity in virus infections involves diverse cellular responses to infections as well as variable infection progressions, which together determine disease outcomes. For example in several life-threatening viral infections immune responses of single cells correlate with disease progression[1–4].

Importantly, the source of cell-to-cell variability in virus infections remains unclear. Genetic diversity in virus populations only partially explains variation in influenza virus infections[5]. Kinetics of poliovirus infections suggest that cellular and viral factors determine heterogeneity at different infection stages[6]. It is also known that cell cycle, size, and neighborhood affect cell-to-cell variation in infections[7,8]. Thus, single-cell studies indicate that infection heterogeneity is a result of a combined action of viral, cellular, and environmental factors.

We explored the possibility that the infection heterogeneity is connected to how the virus is able to alter the cellular state consisting of the physicochemical status and neighborhood of individual cells. As

a model virus, we used herpes simplex virus type 1 (HSV-1) that is a highly prevalent human pathogen[9]. During a lytic infection, the viral nucleocapsid enters the cytoplasm and is transported to the nuclear pores where the double-stranded DNA genome is released into the nucleus, transcribed, and replicated. Based on the sequential expression, the >80 known HSV-1 genes are classified into immediate-early, early, and late[10]. Primary infection of epithelial cells starts a lytic replication cycle of HSV-1 and is followed by life-long latency in peripheral neurons and periodic reactivation. HSV-1 infections are often asymptomatic, but re-entry to a productive replication cycle after reactivation can result in blisters or sores, typically in oral, perioral and ocular sites. Infections of the central nervous system may result in viral meningitis or encephalitis. In immunocompromised individuals, HSV-1 infections can have high morbidity and mortality[11,12].

Single-cell RNA sequencing (scRNA-seq) has revealed high cell-to-cell variability in herpesvirus transcript abundance[13–15], which has been

[1]Institute of Virology, University of Zurich, Zurich, Switzerland. [2]Molecular and Integrative Biosciences Research Programme, University of Helsinki, Helsinki, Finland. [3]Department of Molecular Life Sciences, University of Zurich, Zurich, Switzerland. ✉e-mail: maija.pietilae@uzh.ch; cornel.fraefel@uzh.ch

linked to antiviral responses and activation of developmental programs[13,15]. High expression of genes regulated by nuclear factor erythroid 2-related factor 2 (NRF2) limits HSV-1 infection[15]. Most interferon-stimulated genes (ISGs) are also upregulated in cells with low HSV-1 transcript abundance. However, interferon (IFN) response is activated only in a rare, abortively infected subpopulation. In cells with high HSV-1 transcript abundance, most cellular genes upregulated are involved in developmental processes, including targets of the WNT/β-catenin pathway[13].

It remains, however, unclear how to predict HSV-1 infection progression. Addressing multimodal responses of cells to infection and of viruses to host defense requires multiplexed, spatially resolved quantification of single-cell features at the subcellular, cellular, and multicellular levels that cannot be achieved by scRNA-seq. To maintain spatial information and to measure active protein levels, imaging approaches are required. In this work, we therefore combine single-molecule RNA fluorescence in situ hybridization (smFISH) and iterative indirect immunofluorescence imaging (4i)[16] to uncover heterogeneity in the lytic HSV-1 infection. We use multiplexed high-throughput imaging to analyze hundreds of thousands of individual cells to describe the cellular state landscape in HSV-1-infection and show that it correlates with the infection variability. We find that HSV-1 infection activates phosphatidylinositol 3-kinase (PI3K)/Akt and extracellular signal-regulated kinase 1/2 (ERK1/2) signaling in different subpopulations of cells. Moreover, phosphorylation of RNA polymerase II (RNAP II) is dependent on the infection stage. We also detect loss of processing bodies (PBs) as infection progresses, indicating that they have an antiviral role. Collectively, these four elements are among the cellular factors that correlate with the infection heterogeneity.

## Results

### Both HSV-1 mRNA and protein abundance vary between single cells

To investigate if single-cell heterogeneity is affected by infection time or virus load, we infected HeLa cells with HSV-1 and then quantified cytoplasmic *UL29* and *UL19* transcripts using high-throughput smFISH (Fig. 1a, b). *UL29* encodes infected cell protein 8 (ICP8), a single-stranded DNA-binding protein essential for virus replication and expressed with early kinetics. *UL19* encodes the major capsid protein ICP5, expressed with late kinetics[17]. Synchronized infection resulted in high cell-to-cell variability of both transcripts that was independent of the virus load and present throughout the infection (Fig. 1c and Supplementary 1a). In agreement with their expression kinetics, *UL29* transcript counts were higher at each time point and multiplicity of infection (MOI) compared to *UL19* (Fig. 1a–c).

The coefficient of variation of HSV-1 *UL29* and *UL19* decreased with increasing mean transcript count as has been shown for cellular transcripts, but remained higher compared to that of cellular transcripts (Supplementary Fig. 1b)[18]. Furthermore, variation in the HSV-1 transcript abundance showed significantly lower correlation with a phenotypic state of cells quantified by DNA and protein content, morphology, neighborhood, and cell-cycle phase of single cells when compared to the cellular genes (Supplementary Fig. 1c), suggesting that HSV-1 gene expression has additional sources of variation.

To extend the analysis of infection heterogeneity from mRNAs to proteins and to obtain more information about the cellular state and how it correlates with the infection, we applied multiplexed immunofluorescence imaging to mock and HSV-1-infected HeLa cells at 1.5–12 hpi (smFISH + 4i experiment; Supplementary Fig. 2a). After the 2-plex viral mRNA smFISH, cells were subjected to 36-plex 4i using six antibodies against viral markers, 28 antibodies against cellular markers, a nucleic acid stain, and a total protein stain (Supplementary Fig. 2b). Cellular markers included transcription, signaling, subcellular compartment, cytoskeleton, cell cycle, and antiviral response stains, and we quantified 3136 subcellular, cellular, and multicellular features

by extracting the intensity and texture of the 30 stainings as well as the neighborhood, morphology, and cell-cycle features for each cell (Supplementary Fig. 2c). These single-cell phenotypes form a high-dimensional cellular state space.

Multiplexing allowed observation of several HSV-1 proteins in the same cells, and images revealed highly heterogenous viral protein abundance between single cells (Fig. 1d). Viral markers included immediate-early proteins ICP0, ICP4, and ICP27, early protein ICP8, and late proteins ICP5 and VP16, from which 750 viral single-cell features were extracted (Supplementary Fig. 2d). Cells from the smFISH + 4i experiment were classified infected if they contained at least one incoming virus particle at 1.5 hpi or if they expressed one of the immediate early proteins at 3–12 hpi (Supplementary Fig. 3a–c). Thus, infected cells detected at 1.5 hpi represent the total population of cells with virus attachment or entry while at the later time points only such cells were detected as infected, which had viral gene expression. In addition, uninfected cells were further classified based on their neighborhood: uninfected cells without or with immediate infected neighbors (Supplementary Fig. 3d). Cell counts in different subpopulations at different time points are summarized in Supplementary Fig. 3e. All cell classifications were done using automated computer vision and computational analyses as described in Methods.

Most infections (~80%) originated from a single virus particle, thereby minimizing variation from the number of infecting viruses although not completely excluding it (Supplementary Fig. 3a, b). By 12 hpi a similar fraction of cells was infected as at 1.5 hpi (~30%) (Supplementary Fig. 3c), indicating that most infections were successful. Quantification of HSV-1 protein abundances, however, revealed high cell-to-cell variation (Supplementary Fig. 3f), as was observed for the viral transcripts. Although a synchronized infection protocol was used, different onset times of infection in single cells most likely contribute to the observed heterogeneity in the HSV-1 gene expression.

Next, binomial logistic regression was used on the multivariate dataset to test if single-cell features can predict whether a cell is infected (Fig. 1e). We first used predictive power score (pps) to identify those single-cell features that correlate with uninfected/infected cell type and then used these features as predictors in binomial regression. We identified texture features of seven markers, DEAD (Asp-Glu-Ala-Asp) box helicase 6 (DDX6), phospho-cyclin dependent kinase 9 (p-CDK9), RNAP II phospho-Ser2, RNAP II phospho-Ser5, NRF2, speckled protein 100 kDa (SP100), and total protein staining, that alone detected infection with good prediction accuracy, indicating that infection changed the subcellular distribution of these markers. Furthermore, when all cellular features in the multivariate dataset were combined, even higher prediction accuracy was achieved and the accuracy increased towards the end, highlighting the virus-induced multimodal change of cells. At early times of infection, nevertheless, the multiplexed cellular information could not resolve the infection status, indicating that with respect to the selected markers there was no preexisting cellular state preferred by the virus.

### HSV-1 heterogeneity is independent of cell cycle and size

ScRNA-seq previously connected HSV-1 transcript abundance to the cell cycle in human fibroblasts, but results were conflicting[13,15]. To evaluate the contribution of the cell cycle to HSV-1 infection heterogeneity, we first quantified proliferating cell nuclear antigen (PCNA) in two epithelial cell lines (HeLa and A549) and in fibroblasts (BJ). PCNA is a processivity factor of eukaryotic DNA polymerase delta that is responsible for the synthesis of the lagging strand and is degraded from heterotetramer to heterotrimer to allow cell-cycle progression into the S phase[19,20]. We observed co-localization of PCNA with ICP8-positive viral replication compartments (RCs) (Fig. 2a), as previously described[21]. However, single-cell distributions revealed that HSV-1 infection reduced the mean nuclear intensity of PCNA at late time points in HeLa cells (Fig. 2b and Supplementary 4a) that might reflect

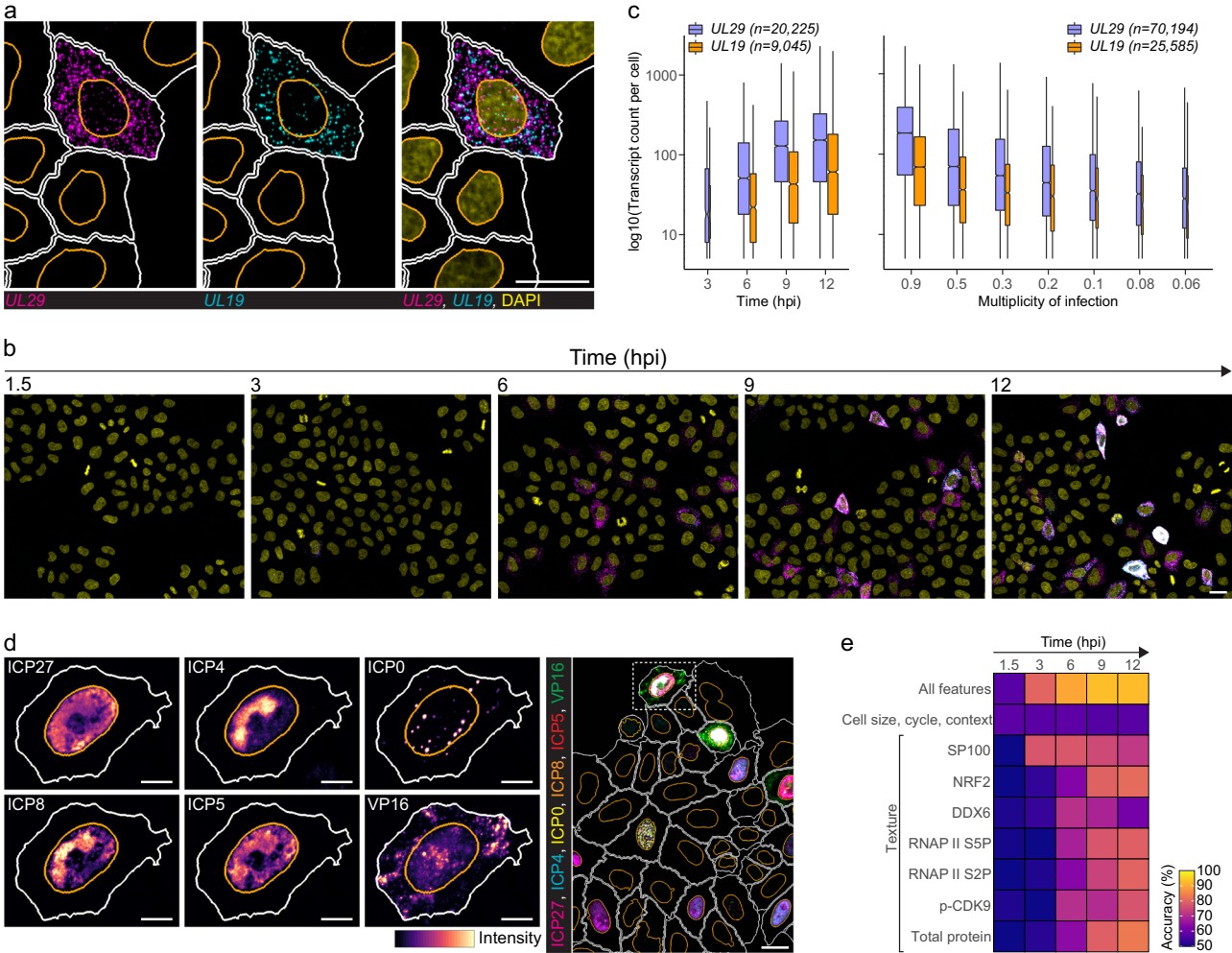

**Fig. 1 | Multiplexed imaging captures heterogenous HSV-1 gene expression between single cells. a**, **b** HeLa cells were infected with HSV-1 multiplicity of infection (MOI) 0.3 and stained for *UL29* (magenta) and *UL19* (cyan) transcripts using smFISH at 1.5–12 hpi. Panel **a**, cells at 9 hpi. Nuclei were stained with DAPI (yellow). Nucleus and cell outlines are indicated in orange and white, respectively. Scale bar, 25 µm. **c** Single-cell transcript counts of *UL29* and *UL19* at MOI 0.3 at 3–12 hpi (left) or at MOIs 0.9-0.06 at 6 hpi (right) in HSV-1-infected HeLa cells. Data were acquired from two independent experiments, and boxplots summarize transcript counts observed in individual replicate wells (left, *n* = 4 and right, *n* = 5). Cell counts are indicated in plots per transcript. Boxplots are as follows: median count of cell population (central mark), 25th percentile (Q1; lower hinge), 75th percentile (Q3;

upper hinge), smallest observation greater than or equal to Q1–1.5*interquartile range (lower whisker), and largest observation less than or equal to Q3 + 1.5*interquartile range (upper whisker). Boxplot width is proportional to the square-roots of the number of cells in each group. **d** Subcellular distribution of viral 4i markers in HeLa cells at 12 hpi. For each marker, a zoom-in image of an example cell is shown next to the composite image (dashed-line). Nucleus and cell outlines as in **a**. Scale bar, 10 µm in single-channel images and 25 µm in composite. **e** Binomial logistic regression to predict whether a cell is infected. For each time point a binomial logistic regression model was trained on cellular features extracted from the smFISH + 4i experiment and used in prediction. Accuracy shown represents mean of four predictions. See also Supplementary Figs. 1–3.

HSV-1-induced changes in the cell cycle. HSV-1 infection resulted in the reduced nuclear PCNA signal also in fibroblasts but not in the other epithelial cell line, A549 (Supplementary Fig. 4a).

Next, we compared abundance of G1-, S-, and G2-cells throughout the infection in HeLa cells (Fig. 2c and Supplementary Fig. 1b, c). Infection initiated in all three cell-cycle phases although G1-cells were less likely infected than S- or G2-phase cells, most likely reflecting the cell size and increased cell area for entry. Infection also progressed in all three cell-cycle phases but by 12 hpi the infected cells contained less S-cells and more G1- and G2-cells than the uninfected ones, in agreement with the HSV-1-induced cell cycle arrests at G1/S and G2/M transitions[22–24]. In A549 cells, HSV-1-infection also reduced the fraction of S-phase cells and increased the fraction of G2-cells (Supplementary Fig. 4d), but the decrease in the S-phase cells was less significant compared to HeLa cells, most likely explaining why no significant decrease in the PCNA intensity was observed in A549 cells. In

fibroblasts the infection-induced changes in the cell-cycle phases also reflected those of HeLa cells (Supplementary Fig. 4e).

Although HSV-1 affected the cell cycle, throughout the infection there was significant overlap in the *UL29* and *UL19* transcript abundance and ICP27 and ICP4 intensity between G1-, S-, and G2-phase cells (Fig. 2d and Supplementary Fig. 5a, b). Besides, cell size showed no correlation with HSV-1 gene expression (Fig. 2e and Supplementary Fig. 5c, d), indicating that HSV-1 gene expression has no scaling with the cell size as has been observed for cellular transcription[25].

### High-dimensional cellular state space explains heterogenous infection

Pairwise Pearson and Spearman correlation analyses showed only low or moderate relationship between cellular and viral features in HeLa cells (Supplementary Fig. 6). However, 18 different cellular markers correlated with HSV-1 gene expression, implying that the high-

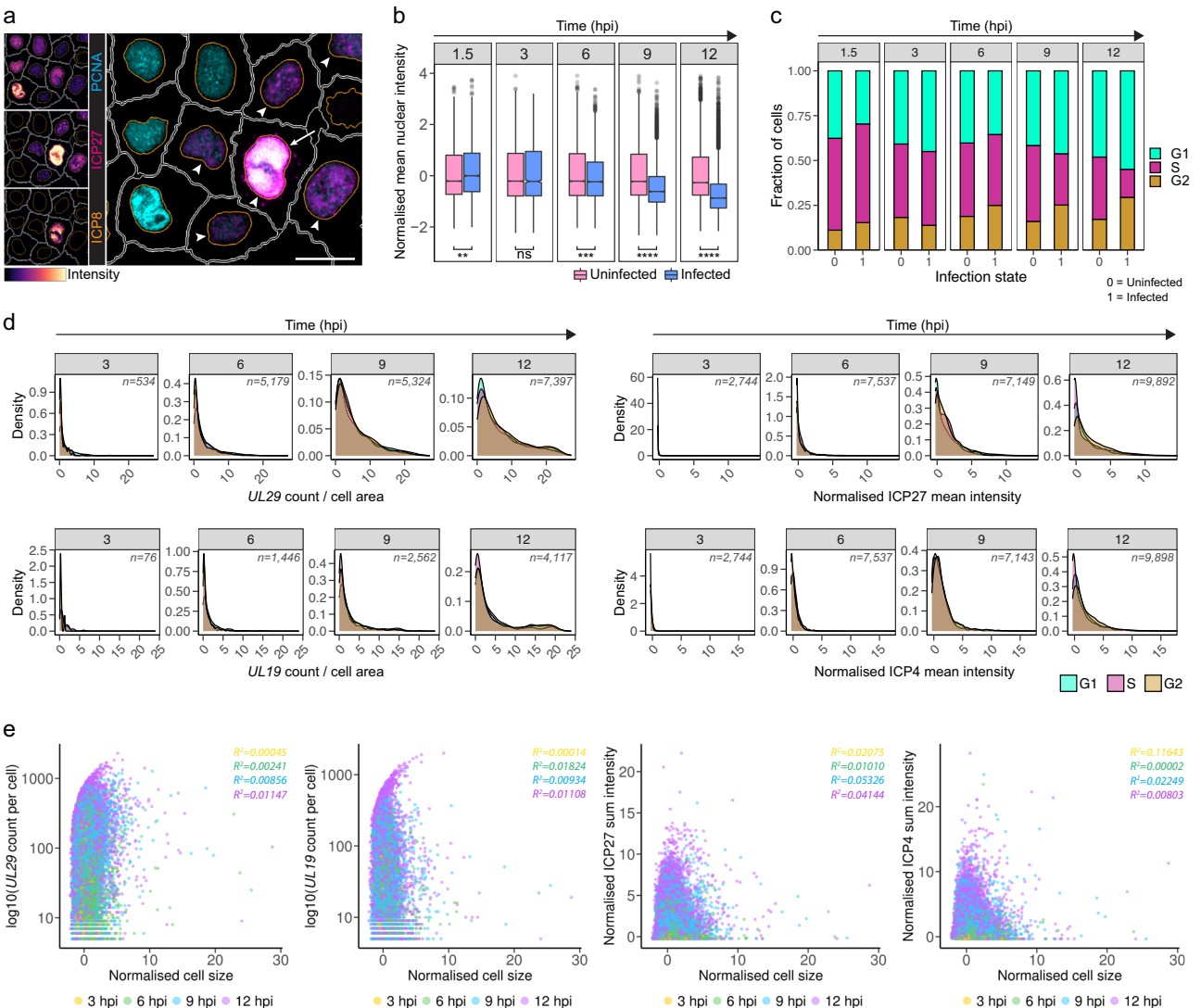

**Fig. 2 | Cell cycle or size cannot explain variation in HSV-1 transcription or translation. a** PCNA, ICP27, and ICP8 staining of HSV-1-infected HeLa cells at 12 hpi. Arrowheads, ICP27 expressing cells. Arrow, an infected cell containing viral RCs with co-localized PCNA. Nucleus and cell outlines as in Fig. 1a. Scale bar, 25 μm. **b** Mean nuclear intensity of PCNA in uninfected or infected HeLa cells from the smFISH + 4i experiment. Data are from four individual replicate wells per time point (1.5 hpi, $n_{uninf.}$ = 18,442 and $n_{inf}$ = 8665; 3 hpi, $n_{uninf.}$ = 26,732 and $n_{inf}$ = 2744; 6 hpi, $n_{uninf.}$ = 22,558 and $n_{inf}$ = 7537; 9 hpi, $n_{uninf.}$ = 28,858 and $n_{inf}$ = 9382; 12 hpi, $n_{uninf.}$ = 25,851 and $n_{inf}$ = 9914 cells). PCNA distributions were compared using pairwise two-sided Kolmogorov–Smirnov (KS) test: *$p < 0.05$, **$p < 0.01$, ***$p < 0.001$, ****$p < 0.0001$, ns not significant. Boxplots definitions as in Fig. 1c. Points, outliers. **c** Cell-cycle phase of uninfected or infected HeLa cells from the smFISH + 4i experiment. Data are from three or four individual replicate wells per time point (1.5 hpi, $n_{uninf.}$ = 18,442 and $n_{inf}$ = 8665; 3 hpi, $n_{uninf.}$ = 26,732 and

$n_{inf}$ = 2744; 6 hpi, $n_{uninf.}$ = 22,558 and $n_{inf}$ = 7537; 9 hpi, $n_{uninf.}$ = 21,920 and $n_{inf}$ = 7155; 12 hpi, $n_{uninf.}$ = 25,851 and $n_{inf}$ = 9914 cells). Statistical comparison is shown in Supplementary Fig. 4b. **d** Single-cell *UL29* and *UL19* transcript counts and ICP27 and ICP4 mean intensities in infected G1, S, and G2-phase HeLa cells. *UL29* and *UL19* transcript counts per cell were normalized by dividing the counts by the cell size and multiplying by 1000. 0.1–99.9th percentiles of marker intensities are shown in the density plots. Data are from three or four individual replicate wells per time point from the smFISH + 4i experiment. Inset: cell counts per time point. **e** Single-cell *UL29* and *UL19* transcript counts and ICP27 and ICP4 sum intensities versus cell size (*UL29*, $n$ = 20,225; *UL19*, $n$ = 9045; ICP27 and ICP4, $n$ = 29,577 cells). Data are from four individual replicate wells per time point from the smFISH + 4i experiment. Inset: $R^2$ of robust linear regression models fit to the single-cell data per time point. See also Supplementary Figs. 4 and 5.

dimensional cellular state space might be able to explain the infection variation.

To better understand multimodal responses of cells to virus infection and the connection to infection heterogeneity, we visualized the high-dimensional cellular state space formed by 3136 single-cell features quantified from the smFISH + 4i experiment as a two-dimensional landscape by embedding the position of single cells within this space in uniform manifold approximation and projection (UMAP)[26]. Cellular features used in UMAPs were first normalized to exclude changes that take place in cells not due to virus infection (see Methods), allowing us to focus on changes in the cellular state during

infection. In addition, all viral features were excluded from this cellular state landscape.

Cells from all time points and all subpopulations, including infected cells, their uninfected neighbors, uninfected cells without immediate infected neighbors, and mock cells, distributed across the landscape, except for a separate cluster formed by the infected cells at 6–12 hpi (Fig. 3a and Supplementary Movie 1). The texture features DDX6, p-CDK9, RNAP II phospho-Ser2, RNAP II phospho-Ser5, NRF2, SP100, and total protein staining, which could predict uninfected/ infected cell fate (Fig. 1e), formed distinct patterns on this cellular state landscape (Supplementary Fig. 7a). Furthermore, some of the

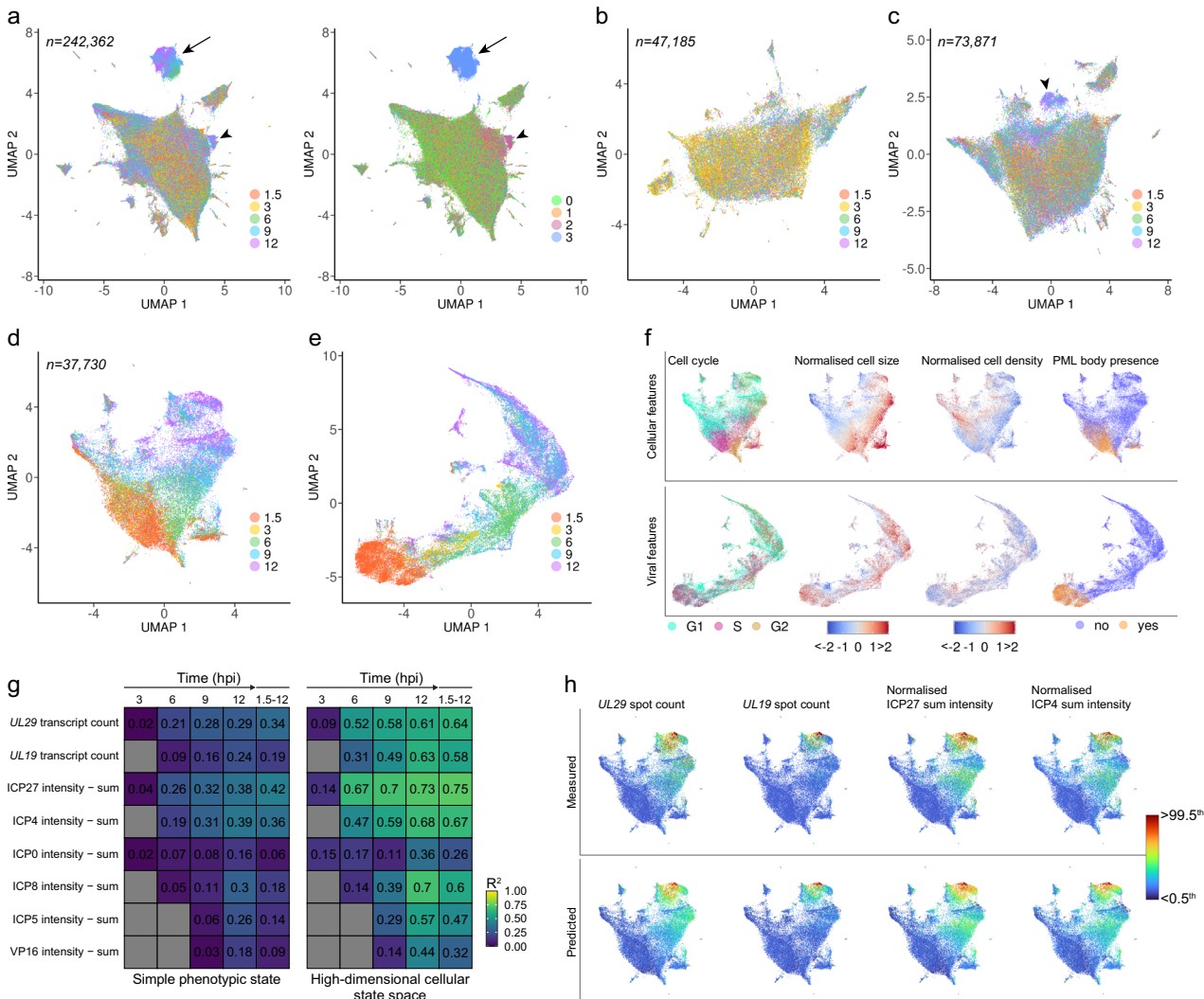

**Fig. 3 | Cellular state landscape correlates with cell-to-cell variation of HSV-1 infection. a** Cellular state landscape of mock- and HSV-1-infected HeLa cells generated from the high-dimensional cellular state space of 3136 cellular features extracted from the smFISH + 4i experiment projected as UMAP. UMAP colored by time point (left; hpi) or by cell subpopulation (right; 0, mock cells; 1, uninfected cells without infected neighbors; 2, uninfected cells with infected neighbors; 3, infected cells). Arrow, a cluster formed by infected cells at 6–12 hpi. Arrowhead, a cluster formed by uninfected cells with infected neighbors at 9 and 12 hpi. Data are from two (mock) or four (HSV-1 infection) replicate wells per time point, and cell count is indicated in the plot. UMAP of uninfected HeLa cells without infected neighbors (**b**), uninfected HeLa cells with infected neighbors (**c**), and infected HeLa cells (**d**), using the same single-cell features as in **a** and colored by time point (hpi). Cell counts are indicated in the plots. **e** UMAP of infected HeLa cells generated from the high-dimensional infection state space (the same cells as in **d**). Position of single

cells within a PC-reduced space of 750 viral features was projected onto a 2-dimensional infection-state landscape using UMAP and colored by time point (hpi). **f** UMAP of infected HeLa cells (the same cells as in **d**), colored by cell cycle, size, or local density, or by PML-body presence. Upper panels, UMAP using the same cellular features as in **a**. Lower panels, UMAP using the same viral features as in **e**. **g** Heatmaps of explained variance. A PC-reduced space of the simple phenotypic state (360 single-cell features) or high-dimensional cellular state space (3,136 single-cell features) was used in MLR to predict viral gene expression in infected HeLa cells. Gray, less than 250 cells expressing the corresponding viral marker. **h** UMAP of infected cells (cellular features) colored by measured or MLR-predicted levels of *UL29*, *UL19*, ICP27, or ICP4 in single cells. Scale bar: lower limit is <0.5th and upper limit is >99.5th percentile of the values. See also Supplementary Figs. 6 and 7 and Movies 1–3.

uninfected cells with infected neighbors enriched in a region of the main cluster at 9 and 12 hpi (Fig. 3a), indicating that besides infected cells, their neighbors experienced HSV-1-induced changes in the cellular state.

Clustering of cells per subpopulation revealed a drastic difference between the uninfected and infected cells (Fig. 3b–d and Supplementary Movies 2 and 3). Uninfected cells, which had infected neighbors, formed a separate cluster at 9 and 12 hpi (arrowhead in Fig. 3c). However, only the infected cells had a time-related pattern on the cellular state landscape, revealing how cells traveled in the high-dimensional cellular state space as infection progressed. This illustrates the multivariate effect the virus infection has on the single-cell

landscape of cellular states. When the viral features extracted from the same cells (Supplementary Fig. 2d) were used to visualize an infection-state landscape, the cell movement through the landscape started earlier and adopted a different shape (Fig. 3e and Supplementary Movie 3). We also observed that at late time points some of the infected cells appeared in the early stages of both landscapes, indicating that they were from a secondary infection (Supplementary Movie 3). Interestingly, the HSV-1 infection-state landscape resembles that of another herpesvirus, human cytomegalovirus, constructed from the single-cell viral transcriptome[27].

Cell cycle, morphology, and cell crowding features distributed in gradients across the cellular state landscape of the infected cells but

differed from the time pattern. On the infection-state landscape, they showed no specific patterns (Fig. 3f). HSV-1 is known to induce ICP0-dependent degradation of promyelocytic leukemia (PML) nuclear bodies early in the infection[28], and PML bodies were present only at the early stage of both landscapes (Fig. 3f).

To assess if the multiplexed phenotypes of single cells can explain the observed cell-to-cell variability in HSV-1 infection, we used the high-dimensional cellular state space to predict HSV-1 gene expression in single cells. Performance of multiple linear regression (MLR) models was quantified by $R^2$ that represents the square of correlation between the predicted and measured values. MLR models explained at least 60% of variance in *UL29*, *UL19*, ICP27, ICP4, and ICP8 expression (Fig. 3g, h), indicating that HSV-1 infection progression correlated with the cellular state landscape. Furthermore, a simple phenotypic state describing DNA and protein content, morphology, neighborhood, and cell cycle of single cells explained much less of the variation in HSV-1 gene expression (Fig. 3g), although these features achieved high explained variance for transcript abundance of cellular genes (Supplementary Fig. 1c).

As expected, linear regression models trained on single features were not able to explain variation in the HSV-1 gene expression, even when the top Pearson correlators, phospho-Akt (p-Akt), DDX6, host cell factor C1 (HCFC1), RNAP II, and RNAP II p-Ser5 (RNAP II S5P), were used as predictors (Supplementary Fig. 7b). However, a combination of these five markers was able to predict, not as much as the high-dimensional cellular state space, but a significant fraction of the variation in HSV-1 infection (Supplementary Fig. 7c). The significance of p-Akt, DDX6, HCFC1, RNAP II, and RNAP II S5P in HSV-1 infection heterogeneity is even more emphasized by a low prediction power of five randomly chosen cellular markers (Supplementary Fig. 7d).

Thus, only by multiplexed quantification of cellular responses to infection and virus-induced changes in the same cells and across thousands of cells can we detect the link between the cellular state and infection heterogeneity. However, although high prediction accuracy uncovers correlation between the multidimensional cellular state space and virus infection variability, it does not reveal the direction of underlying causality. Consequently, the cellular state can define the infection state or vice versa.

### MCUs identify subpopulations among HSV-1-infected cells

To further determine which markers are linked to the infection progression, we measured single-pixel intensities from immunofluorescence images. These can be used to derive detailed maps of subcellular organization by clustering pixels with similar marker-intensity profiles into multiplexed cell units (MCUs)[16]. First, all pixels of a cell and their intensities per marker are extracted resulting in multiplexed pixel profiles. Then, pixels are clustered using these profiles and self-organizing maps (SOMs) resulting in SOM nodes, which are further clustered using Leiden algorithm into MCUs.

In total, 44 MCUs were constructed from the pixel profiles of uninfected and infected HeLa cells, and they formed two main clusters, one enriched for cytoplasmic and the other for nuclear markers (Fig. 4a). Next, we compared how much cell area each MCU occupies in different cell subpopulations (uninfected cells without infected neighbors, uninfected cells with infected neighbors and infected cells), and this revealed HSV-1-induced subcellular re-organization starting at 6 hpi (Fig. 4b).

To further analyze which MCU changes correlated with infection progression, we compared MCU abundance between different cell subpopulations at the 12-h time point (Fig. 4c, d). Compared to uninfected cells, infected cells were enriched for MCUs containing p-Akt and depleted for MCUs containing PCNA, interferon regulatory factor 3 (IRF3), phospho-ERK1/2 (p-ERK), DDX6, and catenin β 1 (CTNNB1) (Fig. 4c). The same was observed when infected cells with high ICP27 expression levels were compared to infected cells with low ICP27 expression (Fig. 4d).

MCUs also enabled us to further study the heterogeneity between infected cells. Clustering of single cells using their MCU abundances identified 13 subpopulations enriched for different MCUs (Fig. 4e). We selected six subpopulations representing high (1 and 13), intermediate (5 and 8), and low (4 and 12) viral gene expression for further analysis. Besides their subcellular organization (Fig. 4f), cells in these subpopulations differed in their morphology (Fig. 4g).

Cells in subpopulations 1 and 13 were enriched for p-Akt-containing MCUs 20 and 25, respectively (Fig. 4e). Cells in subpopulation 13 were smaller and rounder, had higher viral gene expression, and were thus later in infection cycle than those in subpopulation 1 (Fig. 4e, g). Lower levels of ICP0, which is important in the shutdown of host defense[29], might explain why cells in subpopulation 1 did not progress as fast as cells in subpopulation 13. Cells in subpopulations 5 and 8 had relatively high levels of only immediate-early proteins ICP27 and ICP4 and were larger than cells in subpopulations 1 and 13 (Fig. 4e, g), indicating that they were in the early infection stages and originated from a secondary infection.

Subpopulation 4 cells had the highest abundance of cytoplasmic MCU 5 and nuclear MCU 43, in which CTNNB1 and PCNA, respectively, were enriched (Fig. 4e). HSV-1 has been shown to recruit both markers to the viral RCs[13,21]. Thus, the data indicates that HSV-1 failed to hijack these factors to support infection in these cells with low viral gene expression. DDX6-containing MCUs 1 and 3 were the most abundant in subpopulations 12 and 4, respectively. In addition, cells in subpopulation 12 had the highest abundance of MCUs 28 and 41, in which p-ERK and IRF3 were enriched (Fig. 4e). IRF3, p-ERK, and DDX6 all play a role in innate immunity[30,31], and thus our multiplexed data suggest a connection between the host defense and low-viral load in these cells.

### HSV-1 activates signaling in various cell subpopulations

Multiplexed protein maps of the cellular state (Fig. 4) revealed that activation of Akt and ERK pathways is linked to the HSV-1 infection progression. Importantly, such changes in the activity cannot be observed by scRNA-seq approaches. HSV-1 induced phosphorylation of both Akt and ERK at the late time points in HeLa cells but in different cell subpopulations (Fig. 5a, b and Supplementary Fig. 8a–c). p-Akt intensity increased in the infected cells, while p-ERK intensity often increased the most in the uninfected cells next to the infected ones. Interestingly, on the cellular state landscape, elevated p-Akt and p-ERK mean intensities concentrated in the area where infected cells and uninfected neighbors separated (Fig. 5c).

HSV-1 has been reported to either activate or suppress ERK signaling[32–34]. Our single-cell data give more insights into the role of ERK in HSV-1 infection as we observed a strong phenotype in the bystander cells. The higher the levels of viral proteins in the infected HeLa cells, the higher was the mean intensity of p-ERK in the uninfected neighbors (Supplementary Fig. 9a). We therefore asked if this p-ERK increase was induced by IFN signaling to protect the bystander cells, as p-ERK is required in the type I IFN signaling against viruses[35,36]. Notably, the nuclear level of phospho-signal transducer and activator of transcription 1 (p-STAT1) was not increased in the neighbors of the HSV-1-infected cells, nor did Janus kinase (Jak) inhibitor block the observed p-ERK increase, indicating that IFN-induced, Jak1-dependent signaling was not responsible for the ERK activation (Supplementary Fig. 9b, c). Besides, the mean nuclear intensities of IRF3, IRF7, and nuclear factor kappa B (NF-κB), major effectors in pattern recognition receptor (PRR) signaling[37], were not increased in the neighbors (Supplementary Fig. 9d). This also excludes these pathways unless the virus blocked the nuclear transport of these factors after ERK activation. A transcription factor known to restrict HSV-1 infection, NRF2[15], was also unaffected in the neighbors (Supplementary Fig. 9d).

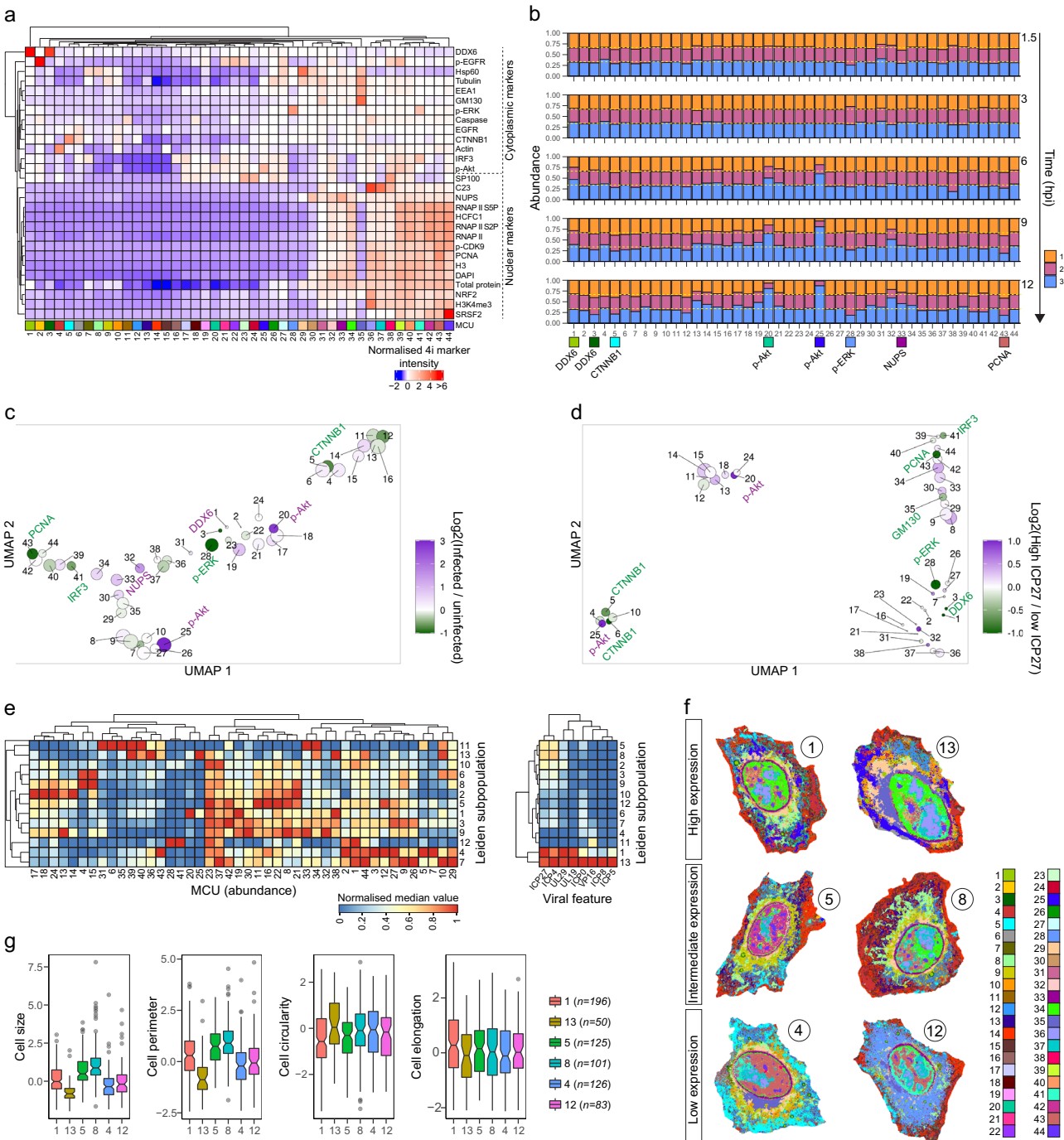

**Fig. 4 | Multiplexed protein maps link cellular changes to infection heterogeneity. a** Heatmap of 4i marker intensities in 44 MCUs from infected and uninfected HeLa cells at 1.5–12 hpi ($n$ = 5500 cells per time point; $n_{total}$ = 27,500 cells). Order of markers and MCUs was calculated by applying hierarchical clustering to the distance matrix created by Euclidean distance measure. **b** Stacked bar plots summarize abundance of MCUs in three cell subpopulations (1, uninfected cells without infected neighbors; 2, uninfected cells with infected neighbors; 3, infected cells). Markers enriched in the selected MCUs are indicated. Dashed yellow lines: 1/3 and 2/3 abundance. **c** UMAP of MCUs colored by their abundance in infected versus uninfected cells at 12 hpi. MCUs were clustered using their spatial interactions from 27,500 infected and uninfected cells (1.5–12 hpi). Markers dominant in the selected MCUs are indicated. Size of each MCU was normalized to the largest MCU. **d** UMAP of MCUs colored by their abundance in high versus low-ICP27 expressing cells at 12 hpi. MCUs were clustered using their spatial interactions from 1520 infected cells at

12 hpi. MCU size on UMAP as in **c**. Low-expressing cells: mean intensity of ICP27 <35th percentile. High-expressing cells: mean intensity of ICP27 >65th percentile. **e** Infected HeLa cells at 12 hpi ($n$ = 1520) were clustered to 13 subpopulations using abundance of each MCU in each cell and Leiden algorithm. Left heatmap: median of MCU abundance in a subpopulation. Right heatmap: median of viral feature in a subpopulation. Medians of subpopulations were standardized and then normalized between 0 and 1. Order of objects was calculated using distance matrix created by Euclidean distance measure and hierarchical clustering. For HSV-1 proteins, sum intensities are shown. For *UL29* and *UL19*, spot counts are shown. **f** Spatial projections of MCUs onto example cells of selected subpopulations. Cells are not drawn to the scale. **g** Cell morphology features of single cells in selected subpopulations. $n$ = cells per subpopulation. Boxplots definitions as in Fig. 1c. Outliers are indicated by points.

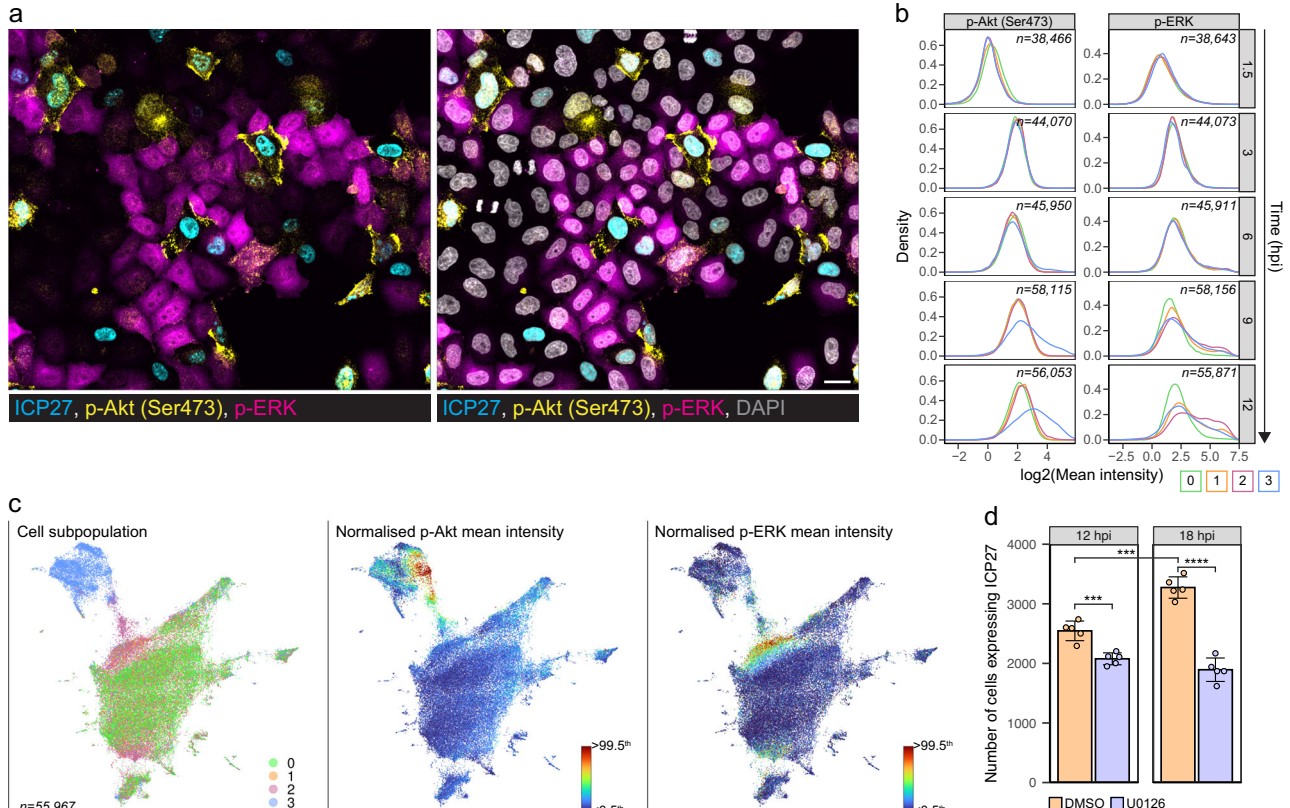

**Fig. 5 | HSV-1 infection activates cell signaling also in neighborhood. a** ICP27, p-Akt (Ser473), p-ERK, and DAPI staining of HSV-1-infected HeLa cells at 12 hpi. Scale bar, 25 μm. **b** Density plots summarize single-cell mean intensities of p-Akt (Ser473) and p-ERK observed in different cell subpopulations from the smFISH + 4i experiment (0, mock cells; 1, uninfected cells without infected neighbors; 2, uninfected cells with infected neighbors; 3, infected cells). Data are from two (mock cells) or four (uninfected and infected cells) individual replicate wells. Cell counts are indicated in plots. **c** UMAP of mock- and HSV-1-infected HeLa cells at 12 hpi colored by cell subpopulations as in **b** (left), p-Akt (Ser473) (middle), or p-ERK (right). UMAP was computed using the same single-cell features as in Fig. 3a. Scale bar: lower limit is <0.5[th] and upper limit is >99.5[th] percentile of the values. Cell count is shown in the first UMAP. **d** Inhibition of ERK phosphorylation. HSV-1-infected HeLa cells were treated with DMSO or U0126 at 1.5 hpi, and ICP27 was detected using immunofluorescence imaging. Data are from two independent experiments and represent mean ± standard deviation among five replicate wells (2 or 3 replicate wells per experiment). Number of ICP27-expressing cells was compared between treatments within a time point or between time points within a treatment using two-sided unpaired two-sample t-test. *$p < 0.05$, **$p < 0.01$, ***$p < 0.001$, ****$p < 0.0001$. 0.1–99.9[th] percentiles of marker intensities are shown in the density plots. See also Supplementary Figs. 8–11.

To study if the ERK activation protects the neighbors from infection, we treated HeLa cells with an upstream mitogen-activated protein kinase (MAPK)/ERK kinase (MEK) inhibitor (MEKi) that blocks ERK phosphorylation[38]. Interestingly, the number of ICP27-expressing cells did not increase from 12 to 18 hpi in the MEKi-treated samples (Fig. 5d), implying that secondary infections were inhibited. We also observed strong ERK activation in the infected cells at 18 hpi (Supplementary Fig. 10a, b). While the uninfected neighbors displayed a diffused cytoplasmic p-ERK signal, in the infected cells p-ERK intensity was either perinuclear or at the plasma membrane (Supplementary Fig. 10b). As newly assembled HSV-1 virions bud through the endoplasmic reticulum/Golgi and plasma membrane[39], our data indicates that HSV-1 activates ERK in the late infection stages to promote virion egress.

Mock-infected A549 cells displayed high p-ERK levels that could be decreased by serum starvation (Supplementary Fig. 11a). However, no neighborhood activation of ERK after HSV-1 infection could be observed, but this may be masked out by the high endogenous p-ERK levels (Supplementary Fig. 11b). Yet, HSV-1 either decreased or increased p-ERK mean intensity in the infected cells, both in serum-starved and non-starved conditions (Supplementary Fig. 11b, c). As observed in HeLa cells, p-ERK intensity was perinuclear in the infected A549 cells (Supplementary Fig. 11c). Mock-infected fibroblasts also had high endogenous p-ERK levels that were reduced by serum starvation

(Supplementary Fig. 11d–f). Similar to HeLa and A549 cells, the infected fibroblasts displayed aggregated p-ERK signal that was perinuclear or at the plasma membrane (Supplementary Fig. 11g). Thus, in all three cell lines HSV-1 infection activated ERK that resulted in different subcellular distribution of p-ERK compared to the uninfected cells.

Besides HeLa cells, HSV-1 infection increased p-Akt levels in the infected A549 and BJ cells (Supplementary Fig. 12). Akt phosphorylation in HSV-1-infected cells has been proposed to enhance translation by activating mechanistic target of rapamycin complex 1 (mTORC1) and to protect cells from apoptosis before these functions of Akt are overtaken by Us3 kinase, a late HSV-1 protein[40–43]. Thus, we first compared mean intensities of p-Akt and cleaved caspase 3, a hallmark of apoptotic cells[44]. Cleaved caspase 3 mean intensity increased only in the infected cells with low p-Akt levels (Fig. 6a). To further explore the role of Akt, we treated cells with a PI3K inhibitor, LY294002. Inhibitor treatment reduced HSV-1-induced phosphorylation of both Akt Ser473 and Thr308 as well as expression of HSV-1 early and late proteins (Fig. 6b and Supplementary 13a), suggesting that Akt activation was also required in the HSV-1 translation. mTORC1 activation by Akt leads to phosphorylation of eukaryotic translation initiation factor 4E (eIF4E)-binding protein (4E-BP) and release from eiF4E and subsequently translation initiation[45]. Phosphorylated 4E-BP is then degraded[46]. HSV-1 infection reduced the 4E-BP mean intensity, but the LY294002-treatment restored 4E-BP levels in the infected cells closer

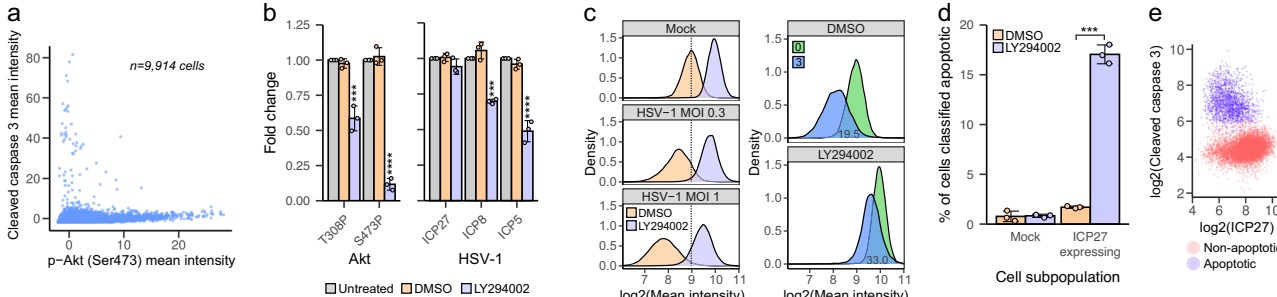

**Fig. 6 | HSV-1 activates Akt to suppress apoptosis and to enhance translation.**
**a** Cleaved caspase 3 as a function of p-Akt (Ser473) in the infected HeLa cells at 12 hpi from the smFISH + 4i experiment ($n = 4$ wells). **b** HeLa cells were infected with HSV-1 (MOI 1) and p-Akt and ICPs were detected in untreated and DMSO- or LY294002-treated cells at 12 hpi using Western blotting. Treated samples were compared to the untreated samples to give a fold change. Data represent mean ± standard deviation from three independent experiments. Treatments were compared by one-way ANOVA with Tukey's multiple comparison post-test relative to untreated samples. **c** Single-cell intensity of 4E-BP in mock or HSV-1-infected in HeLa cells after DMSO or LY294002 treatment, from three replicate wells (mock $n_{DMSO} = 34,389$, $n_{LY294002} = 23,533$; HSV-1 MOI 0.3, $n_{DMSO} = 28,067$, $n_{LY294002} = 19,455$; HSV-1 MOI 1 $n_{DMSO} = 25,360$, $n_{LY294002} = 20,700$ cells). 4E-BP and ICP27 were detected by immunofluorescence imaging at 12 hpi. Right: mock cells (0) and ICP-expressing cells from HSV-1 infection using MOI 0.3 (3). The overlapped estimated area of two distributions is indicated as a mean percentage from

two biologically independent experiments ($n = 3$ wells per experiment). Data from a biological replicate are shown in Supplementary Fig. 13b. **d** Apoptotic cells in mock or HSV-1-infected HeLa cells (MOI 0.3) treated with DMSO or LY294002. Cleaved caspase 3 and ICP27 were detected by immunofluorescence imaging at 12 hpi. Mock cells ($n_{DMSO} = 32,328$ and $n_{LY294002} = 24,073$ cells) and ICP27-expressing cells from HSV-1 infection ($n_{DMSO} = 13,376$ and $n_{LY294002} = 10,513$ cells). Data are from one experiment and represent mean ± standard deviation among three replicate wells (Supplementary Fig. 13c shows data from a biologically independent experiment). Percentage of apoptotic cells was compared between the treatments by one-sided unpaired two-sample t-test. **e** Mean intensity of cleaved caspase 3 as a function of ICP27 in the LY294002-treated, ICP27-expressing HeLa cells ($n_{non-apoptotic} = 8713$ and $n_{apoptotic} = 1800$ cells). Data are from the same experiment as in **d**. In **b** and **d**: *$p < 0.05$, **$p < 0.01$, ***$p < 0.001$, ****$p < 0.0001$. 0.1–99.9th percentiles of marker intensities are shown in the density plots. See also Supplementary Fig. 13.

to those observed in the mock cells (Fig. 6c and Supplementary 13b), indicating that Akt phosphorylation is required in the infected cells to inactivate 4E-BP but that HSV-1 has additional means to enhance 4E-BP degradation and thus translation, most likely by Us3 kinase[41]. Besides affecting translation, the LY294002-treatment increased the fraction of apoptotic cells significantly among the infected cells (Fig. 6d, e and Supplementary 13c, d), confirming the role of Akt in the inhibition of HSV-1-induced apoptosis.

## HSV-1 induces loss of processing bodies

HSV-1 modifies multiple liquid-phase separated compartments of the host cells, including nuclear compartments such as splicing speckles, nucleoli, and PML bodies[47–49]. Here, we observed that HSV-1 infection also changed cytoplasmic liquid-phase separated compartments. RNA helicase DDX6 is a component of processing bodies (PBs) that are ribonucleoprotein granules and most likely function in translational repression and/or mRNA decay[50]. Besides RNA metabolism, PBs play an important role in innate immunity[31,51,52]. MCU analysis implied that DDX6 has an antiviral role in HSV-1 infection (Fig. 4), and we observed that the mean intensity of DDX6 decreased in HSV-1-infected HeLa cells at the late stages of infection (Fig. 7a and Supplementary Fig. 14a).

At 12 hpi and MOI 0.3, HSV-1-infected HeLa and A549 cells typically had few enlarged PBs or no PBs (Fig. 7b). As a control we first quantified nuclear PML bodies that disappeared from the infected HeLa cells by 3 hpi (Fig. 7c). Segmentation and quantification of PBs in HeLa cells revealed that their number decreased as infection progressed, and at the same time their mean area increased (Fig. 7c and Supplementary 14b), indicating that PBs fused before they disappeared. At 12 hpi, 19.4% of the infected HeLa cells had no PBs while only 5.7% of the uninfected cells (mean of two independent experiments) were lacking PBs. DDX6 mean intensity and PB count decreased the most in the G1- and G2-phase HeLa cells (Fig. 7d). In addition, the higher the ICP27 levels were, the lower the PB count was in HeLa cells (Fig. 7e), and the pattern of DDX6 texture followed the infection progression on the cellular state landscape (Fig. 7f), both supporting the hypothesis that infection resulted in the loss of PBs.

Quantification of PBs in A549 cells revealed the same as in HeLa cells (Fig. 7g). However, the mean intensity of DDX6 decreased only in

some HSV-1-infected A549 cells while in some it increased (Supplementary Fig. 14c). A further DDX6-phenotype was observed and was dominant when cells were infected with MOI 1: DDX6 formed a perinuclear aggregate explaining the increased DDX6 mean intensity (Supplementary Fig. 14d). This phenotype was also observed in HeLa cells infected with MOI 1 but was not as common as in A549 cells (Supplementary Fig. 14e). In human fibroblasts this perinuclear aggregate was readily detectible already at MOI 0.3 (Supplementary Fig. 14f). Thus, in all three cell lines HSV-1 infection drastically changed PBs and localization of DDX6 and this was dependent on the virus load. HSV-1-induced PB formation has previously been reported in HeLa cells[53], but it was not observed here (Fig. 7c).

## RNAP II S5P levels fluctuate in infected cells

HSV-1 uses cellular RNAP II and its regulatory factors for transcription[54], and our correlation analyses between HSV-1 gene expression and cellular factors pointed out total RNAP II, RNAP II S5P and HCFC1 (Supplementary Figs. 6 and 7). Phosphorylation of the carboxy-terminal domain (CTD) of the largest subunit of RNAP II, B1, regulates and coordinates activity of the polymerase. CTD is composed of tandem heptad repeats, and phosphorylation of Ser5 positions is a hallmark of initiation. After the promoter-proximal pausing, transition of RNAP II from initiation to elongation requires phosphorylation of Ser2 positions (RNAP II S2P)[55]. It is known that HSV-1 infection eventually results in the depletion of phosphorylation at Ser2 positions[56–59].

The multiplexing approach allowed us to spatially quantify total RNAP II, RNAP II S5P and S2P, and HCFC1, a transcriptional coregulator essential for HSV-1 immediate-early gene expression[60], in the same cells. To our surprise, we observed reduction of phosphorylation at both Ser5 and Ser2 positions in HeLa cells as infection progressed (Fig. 8a). Mean nuclear intensity of RNAP II S5P and S2P decreased starting at 6 and 9 hpi, respectively, indicating that HSV-1 inhibited initiation and subsequently elongation. The nuclear levels of HCFC1 also decreased at late time points in the infected cells. Reduction of phosphorylation at Ser5 and Ser2 positions was verified in HeLa and A549 cells at 12 hpi (Supplementary Fig. 15a, b). However, in human fibroblasts no decrease of RNAP II S5P and S2P nuclear mean intensities

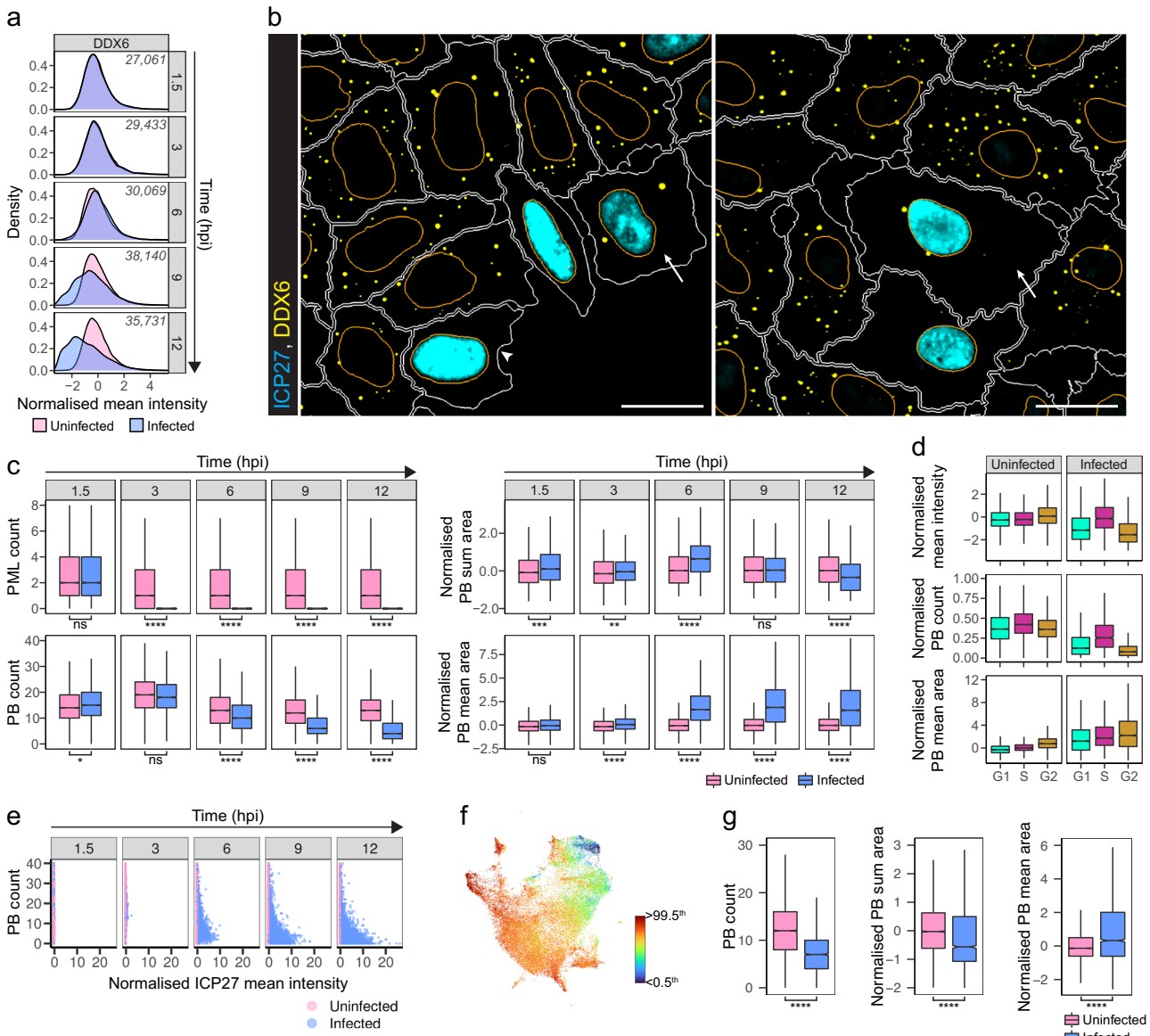

**Fig. 7 | HSV-1 infection results in enlargement and then disappearance of processing bodies. a** Distributions of DDX6 intensity in uninfected and HSV-1-infected HeLa cells from four individual replicate wells in the smFISH + 4i experiment. 0.1–99.9th percentiles of marker intensities are shown in the density plots. Inset: cell counts per time point. **b** DDX6 and ICP27 staining of HSV-1-infected HeLa (left) or A549 (right) cells at 12 hpi. Arrowhead, ICP27-expressing cell without PBs. Arrows, ICP27-expressing cell with two enlarged PBs. Nucleus and cell outlines as in Fig. 1a. Scale bar, 25 μm. **c** Boxplots summarize single-cell PML-body and PB counts as well as PB sum and mean area in single uninfected and infected HeLa cells from the smFISH + 4i experiment (*n* = 4 replicate wells). Distributions were compared using pairwise two-sided KS test. Boxplots definitions as in Fig. 1c. Outliers are omitted for clarity. **d** DDX6 intensity, PB count, and PB area in single uninfected and infected HeLa cells at 12 hpi, per cell-cycle phase, from the smFISH + 4i experiment

(*n* = 4 replicate wells). PB count was normalized by cell area and multiplied by 1000. Boxplots definitions as in Fig. 1c. Outliers are omitted for clarity. **e** PB count as a function of ICP27 intensity in single uninfected and infected HeLa cells. Cells with a PB count >40 are not shown. **f** UMAP of infected HeLa cells (as in Fig. 3d) colored by normalized Haralick sum variance (Kernel size: 2) texture of DDX6 in cytoplasm. Scale bar: lower limit is <0.5th and upper limit is >99.5th percentile of the values. **g** Boxplots summarize single-cell PB count, sum and mean area in uninfected and infected A549 cells at 12 hpi. Distributions were compared using pairwise two-sided KS test. Data are from two independent experiments (*n* = 2-3 wells per experiment). *n* = 9878 uninfected and *n* = 2858 infected cells. Boxplots definitions as in Fig. 1c. Outliers are omitted for clarity. In **c** and **g**: *\**p* < 0.05, **\**p* < 0.01, ***\**p* < 0.001, ****\**p* < 0.0001, ns not significant. See also Supplementary Fig. 14.

was observed when infected cells were compared to uninfected ones (Supplementary Fig. 15c).

Although transcriptional marker intensities decreased in the infected HeLa cells, total RNAP II, RNAP II S5P, and HCFC1 enriched in the viral RCs positive for HSV-1 ICP8 (Fig. 8b). To compare phosphorylation status of RNAP II at different infection stages (Fig. 8c), we first classified infected cells to RC-containing cells (class V) and those without RCs, and the later cells were then clustered to four classes (classes I–IV) based on their expression of HSV-1 immediate early

proteins ICP4 and ICP27 as those were the most abundantly expressed HSV-1 proteins studied here and their expression increased until 12 hpi (Supplementary Fig. 3f). Thus, classes I–V represent infected cells from early to late stages (Fig. 8d). RNAP II S5P intensity decreased when infection progressed but then again increased when cells reached the latest stages (Fig. 8c, e). RNAP II S2P and HCFC1 amounts consistently decreased as the infection progressed (Fig. 8c, e), revealing different requirements for RNAP II S5P, RNAP II S2P, and HCFC1 at different infection stages. HSV-1-induced changes in the

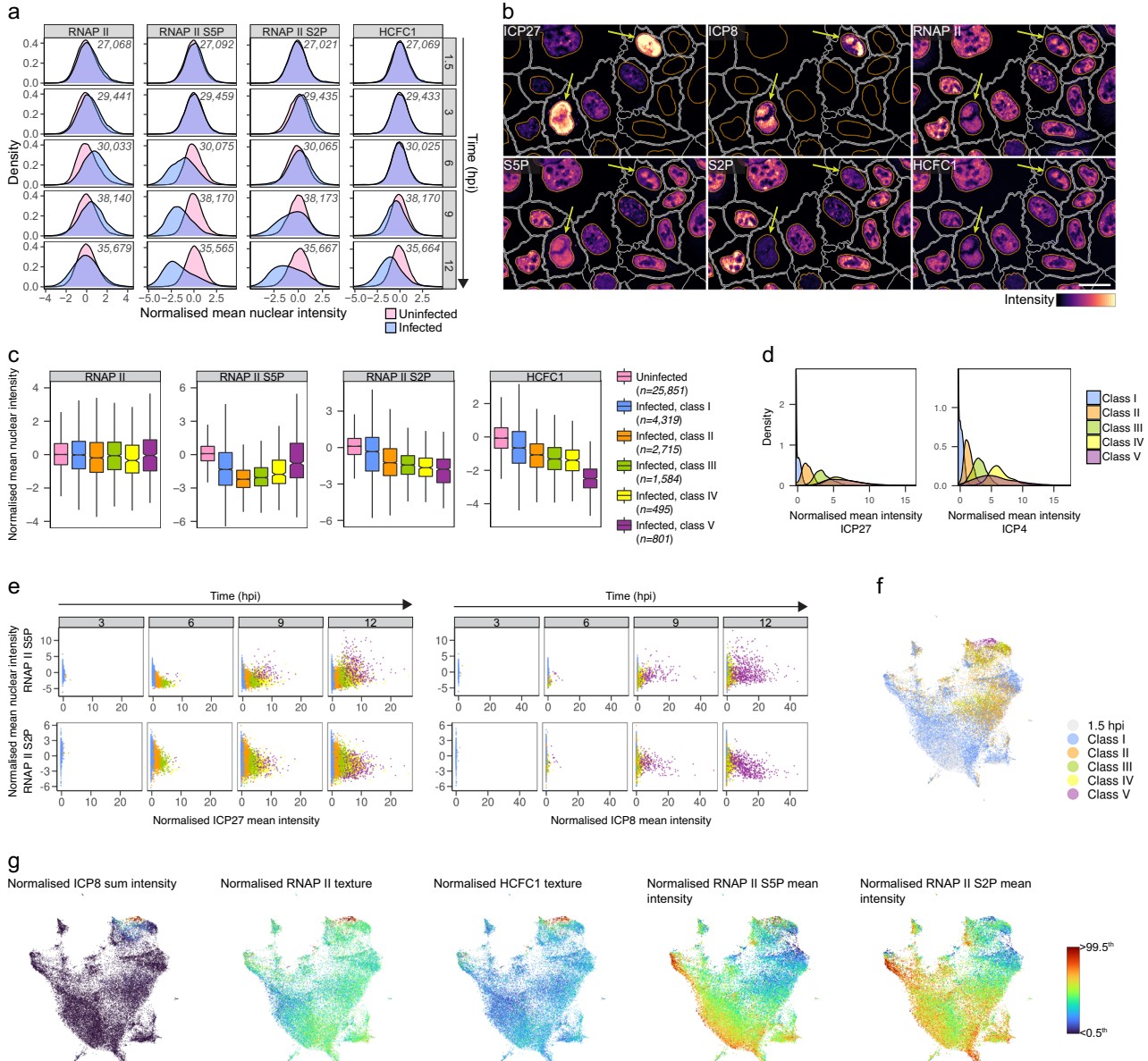

**Fig. 8 | Multiplexing reveals that RNAP II phosphorylation fluctuates during HSV-1 infection. a** Distributions of mean nuclear intensities of total RNAP II, RNAP II S5P, RNAP II S2P, and HCFC1 in uninfected and HSV-1-infected HeLa cells from four individual replicate wells in the smFISH + 4i experiment. 0.1–99.9th percentiles of marker intensities are shown in the density plots. Inset: cell counts per time point. **b** ICP27, ICP8, total RNAP II, RNAP II S5P, RNAP II S2P, and HCFC1 staining of HSV-1-infected HeLa cells at 12 hpi. Arrows, infected cells containing viral replication compartments (RCs) with RNAP II, RNAP II S5P, and HCFC1 co-localized. Nucleus and cell outlines as in Fig. 1a. Scale bar, 25 μm. **c** Mean nuclear intensities of total RNAP II, RNAP II S5P, RNAP II S2P, and HCFC1 in uninfected and infected HeLa cells at 12 hpi. Infected cells without RCs were clustered into four classes (I–IV) based on their mean intensity of ICP27 and ICP4. Class V represents infected cells

that contain at least one ICP8-positive RC. Data are from four individual replicate wells in the smFISH + 4i experiment, and cell counts are indicated next to plots. Boxplots definitions as in Fig. 1c. Outliers are omitted for clarity. **d** Single-cell mean intensity of ICP27 and ICP4 in classes I–V of infected HeLa cells, at 12 hpi. **e** Single-cell mean nuclear intensity of RNAP II S5P or S2P as a function of mean intensity of ICP27 or ICP8 in infected HeLa cells at 3–12 hpi. Classes as in **c**. **f** UMAP of infected HeLa cells (as in Fig. 3d) colored by classes as in **c**. Cells at 1.5 hpi were not included in the clustering and are colored gray. **g** UMAP of infected HeLa cells (as in Fig. 3d) colored by ICP8 sum intensity, Haralick sum entropy (Kernel size: 10) texture of RNAP II in nucleus, Haralick correlation (Kernel size: 2) texture of HCFC1 in nucleus, and mean nuclear intensities of RNAP II S5P and S2P. Scale bar: lower limit is <0.5th and upper limit is >99.5th percentile of the values. See also Supplementary Fig. 15.

transcriptional markers were also visible on the cellular state landscape (Fig. 8f, g).

We next compared mean nuclear intensities of RNAP II S5P and S2P in HeLa, A549 and BJ cells between cells expressing ICP8, an early protein, with and without RCs (Supplementary Fig. 15d–f). In all three cell lines, RC-containing cells had higher RNAP II S5P and lower RNAP II S2P intensities than those without RCs, supporting a conclusion that RNAP II phosphorylation status is dependent on the infection stage. As

in HeLa cells, RNAP II S5P, but not S2P, enriched in the viral RCs in A549 and BJ cells (Supplementary Fig. 15g, h).

## Discussion

Using HSV-1 as a model we show that virus infection changes the cellular state landscape and at the same time, the high-dimensional cellular state space can predict infection heterogeneity (Fig. 3). Consequently, multimodal responses of cells to infection and of

viruses to host are an important source of information to predict infection variability and eventually disease outcomes.

Multiplexed cellular protein maps (Fig. 4) allowed us to classify infected cells into three main categories: 1) cells with low HSV-1 expression from abortive infections, 2) cells with intermediate HSV-1 expression from secondary infections, and 3) cells with high HSV-1 expression from successful primary infections. Cytoplasmic p-ERK, CTNNB1, and DDX6 as well as nuclear IRF3 and PCNA are key identifiers of low-expressing cells. High viral gene expression is achieved in cells that activate Akt, and we show that HSV-1 requires p-Akt to enhance translation of its proteins as well as to block infection-induced apoptosis (Fig. 6 and Supplementary 13). Suppression of ERK activation is important for infection to progress, but the final steps of the infection cycle seem to activate ERK signaling (Fig. 5 and Supplementary Figs. 10 and 11). Controversially, of these cellular factors scRNA-seq could only capture activation of the WNT/β-catenin pathway in cells with high HSV-1 transcript abundance[13].

Importance of spatially resolved single-cell information is further strengthened by our finding that among HeLa cells HSV-1 infection activates ERK in the neighboring cells (Fig. 5). ERK phosphorylation was independent on Jak1-mediated IFN signaling and did not result in activation of PRR transcription factors (Supplementary Fig. 9), indicating that instead of innate immunity, p-ERK rather regulates cell survival, proliferation, or apoptosis of the neighbors by other means.

Besides signaling responses, this study uncovers the role of PBs in HSV-1 infection and suggests that infection progression depends on degradation or aggregation of DDX6 and loss of PBs (Figs. 4 and 7 and Supplementary 14). During lytic infection, Kaposi's sarcoma-associated herpesvirus (KSHV) disrupts PBs to promote its replication, and it has been shown that expression of KSHV ORF57 or its HSV-1 homolog, ICP27, induces PB loss[61]. This agrees with our finding that among the HSV-1 markers studied here, the PB count shows the highest correlation with ICP27 (Supplementary Fig. 6). Enterovirus 71, an RNA virus, encodes a protease that degrades DDX6 to suppress antiviral signaling mediated by retinoic acid-inducible gene I (RIG-I)[31]. As RIG-I can also sense RNA species produced during HSV-1 infection[62,63], our observations suggest that HSV-1 degrades DDX6 to evade host immune responses.

Using multiplexed immunofluorescence imaging, we also identified another cellular factor, RNAP II, that explains HSV-1 single-cell heterogeneity together with p-Akt, DDX6, and HCFC1 (Supplementary Fig. 7). Specifically, we discover that RNAP II concentration is stable in the infected cells, but its phosphorylation status fluctuates and depends on cell subpopulation (Fig. 8 and Supplementary 15). Previously it has been reported that HSV-1 induces loss of RNAP II S2P[56]. However, in our time-course, multiplexed quantification of RNAP II activity in HeLa cells, we observe that the infection results in the reduction of both RNAP II S5P and S2P, but then RNAP II S5P recovers (Fig. 8). The reduction of both phosphoforms was also observed in another epithelial cell line, A549, but not in fibroblasts (Supplementary Fig. 15). However, in all three cell lines the initiation active form of RNAP, S5P, enriched in the HSV-1 RCs, but the elongation active form, S2P, was depleted. These HSV-1-induced changes in the RNAP II activity must reflect both the shutdown of cellular transcription[64,65] and regulation of viral transcription. It has been suggested that HSV-1 transcription is not subjected to the promoter-proximal pausing relieved by S2P[56]. However, as both RNAP S5P and S2P are known to bind to the HSV-1 genome and inhibition of Ser2 phosphorylation impairs viral transcription[66,67], our data suggest that this is true only when viral genes are transcribed within the HSV-1 RCs (Fig. 8 and Supplementary 15). RNAP II is recruited to the HSV-1 RCs through nonspecific binding of RNAP II to the viral DNA[68], and thus there may be no synchronized action of RNAP II enabled by the promoter proximal pausing and consequently no need for S2P.

The observed fluctuations in RNAP II phosphorylation could also be partially explained by a model of cellular mRNA concentration homeostasis. Negative feedback from mRNA concentration on the RNAP II activity is proposed to prevent transition from pausing to elongation, and this would result in a greater loss of S2P than S5P[25]. Interestingly, inhibition of transcription in HSV-1-infected cells by actinomycin D prevents the loss of S2P[69].

Collectively, multidimensional spatial readouts of cellular phenotypes are a necessity to understand heterogenous responses in infections. Although the multiplexed immunofluorescence imaging is limited in the number of markers compared to scRNA-seq and the choice of markers has an impact on conclusions, our approach allows spatial quantification of tens of features per marker as well as capturing cell morphology and neighborhood features, increasing the dimensionality. Importantly, we identified multiple cellular factors that explain single-cell variation in human herpesvirus infection and that have not been identified in scRNA-seq approaches. Further studies are required to reveal the direction of causality between the cellular factors and viral gene expression. However, the multiplexed imaging enables quantification of active host and virus responses and could enhance for example screening of targets for antivirals.

## Methods
### Cell lines and viruses
HeLa Kyoto cells (cervical cancer, human, female) are a single-cell clone and have been authenticated by karyotyping (Supplementary Data 1)[70]. HeLa, A549 (lung cancer, human, male) and BJ (normal foreskin fibroblasts, human, male) cells (Supplementary Data 1) were cultured in low-glucose Dulbecco's Modified Eagle's Medium (DMEM), supplemented with 10% (v/v) fetal bovine serum (FBS). Vero (kidney tissue, African Green Monkey) was maintained in low-glucose DMEM supplemented with 10% FBS, penicillin (100 units/mL), streptomycin (100 μg/mL), and amphotericin B (250 ng/mL). All cell lines were grown at 37 °C and 5% $CO_2$. Wild-type herpes simplex virus 1 (HSV-1) strain F was propagated in Vero cells (Supplementary Data 1). Details of cell culture media are indicated in Supplementary Data 1.

### HSV-1 purification
HSV-1 stocks were produced in Vero cells as previously described[71] with some modifications. Briefly, $4 \times 10^6$ cells were seeded in 150-cm² culture flasks and grown overnight at 37 °C and 5% $CO_2$. Next day, cells were infected with HSV-1 (MOI 0.1) by incubating with a virus inoculum in serum-free DMEM for 30 min at 4 °C and 60 min at 37 °C. Unbound virus was removed, and cells were incubated in DMEM supplemented with 2% (v/v) FBS at 37 °C and 5% $CO_2$ until complete cytopathic effect was observed (at ~48 hpi). Cells were collected into the medium by scraping and sonicated twice for 30 s in a water bath with a 10-s incubation on ice in between. Sonication cycles were followed by three freezing-and-thawing cycles (liquid nitrogen and 37 °C-water bath) and by two sonication cycles as above. Cell debris was removed by centrifugation (Thermo Scientific Multifuge X3R, 2000 × g, 5 min, 4 °C), and virus suspension was purified through a 30% (w/v) sucrose cushion in TNE buffer (10 mM Tris-HCl pH 8.0, 100 mM NaCl, 1 mM EDTA pH 8.0) by ultracentrifugation (AH629, 140,000 × g, 2 h, 4 °C). The virus pellet was resuspended in TNE buffer and sonicated three times for 10 s in a water bath with a 10-s incubation on ice in between. The purified virus was stored in aliquots at −80 °C.

The purified HSV-1 was titrated in HeLa, A549 and BJ cells using immunofluorescence imaging and an antibody detecting ICP27 (Supplementary Data 1).

### Synchronized virus infection
For single-molecule RNA fluorescence in situ hybridization (smFISH) and immunofluorescence imaging 3500 HeLa, 4000 A549 or 2500 BJ

cells were seeded per well in uncoated Greiner µClear plastic-bottom 96-well plates (Supplementary Data 1). HeLa cells were grown for ~48 h and A549 and BJ cells for ~72 h at 37 °C and 5% $CO_2$ to reach ~80% confluency. Cells were then incubated for 20 min at 4 °C, washed three times with cold serum-free DMEM, and infected with the purified HSV-1 diluted in serum-free DMEM using MOI 0.3, unless otherwise stated in the figure legends. Before the virus was diluted, it was sonicated for 10 s in a water bath. To allow virus adsorption, cells were incubated with the virus for 30 min at 4 °C, and then unbound virus was removed by washing cells three times with warm DMEM supplemented with 10% (v/v) FBS. Cells were then incubated 60 min at 37 °C to allow virus internalization. Non-internalized virus was removed by washing cells twice with acid buffer (40 mM Na citrate, 135 mM NaCl, 10 mM KCl, pH 3.0) and incubating in the acid buffer for 1 min. Cells were then washed four times with warm DMEM supplemented with 10% (v/v) FBS and subsequently grown at 37 °C before fixation and staining. For A549 cells, no acid wash was performed to prevent cell damage. If serum starvation was performed, cells were incubated in DMEM supplemented with 0.2% (v/v) FBS for 12 h before infection, and after the virus adsorption in serum-free DMEM, cells were incubated in DMEM supplemented with 0.2% (v/v) FBS.

For Western blotting 30,000 HeLa cells were seeded per well in uncoated TPP tissue culture 12-well plates. The infection protocol was the same as above except, that cells were washed only once before infection and twice after the 30 min adsorption step and no acid wash was performed. At 12 hpi, cells were washed twice with ice-cold PBS and collected in 2× Laemmli sample buffer (4% (w/v) SDS, 10% (v/v) 2-mercaptoethanol, 20% (v/v) glycerol, 0.1% (w/v) bromophenol blue, 100 mM Tris-HCl pH 6.8) and incubated for 5 min at 99 °C.

### Chemical treatments

MEK inhibitor U0126 and PI3K inhibitor LY294002 (Supplementary Data 1) were dissolved in DMSO at a concentration of 50 mM and added to cells at 1.5 hpi at a final concentration of 50 µM. JAK1 inhibitor (Supplementary Data 1) was dissolved in DMSO at a concentration of 10 mM and used at 10 µM. Specifically, cells were pre-treated with JAK1 inhibitor for 3 h before infection and then JAK1 inhibitor was added again at 1.5 hpi. Human Interferon-γ (hIFN-γ) (Supplementary Data 1) was dissolved at a concentration of 0.1 mg/mL in water supplemented with 0.1% (w/v) bovine serum albumin (BSA) (Supplementary Data 1). Cells were treated with hIFN-γ (100 ng/mL) for 30 min before fixation.

### Western blotting

Proteins were separated by sodium dodecyl sulfate–polyacrylamide gel electrophoresis with 4% (w/v) acrylamide in the stacking gel and 10 or 12% (w/v) acrylamide in the separation gel. After electrophoresis, proteins were transferred to nitrocellulose blotting membrane (Supplementary Data 1), which was subsequently blocked with 5% (w/v) milk in PBS for 1 h at RT and incubated with primary antibodies (Supplementary Data 1) overnight at 4 °C. After the overnight incubation, membranes were washed three times with 0.3% (v/v) Tween 20 (Supplementary Data 1) in PBS and then incubated with secondary antibodies (Supplementary Data 1) for 1 h at RT. Antibodies were diluted in PBS supplemented with 2.5% (w/v) milk and 0.3% (v/v) Tween 20. Membranes were washed three times with 0.3% (v/v) Tween 20 in PBS, once with PBS, and subsequently scanned with Odyssey system (LI-COR Biotechnology).

Protein band intensities were quantified using Fiji (Supplementary Data 1) as the area under the curve of each band. All cellular and viral markers were normalized by dividing their intensity by the corresponding β-actin intensity. Fold change was calculated by comparing normalized protein intensity in the DMSO- or LY294002-treated samples to the untreated samples. Western blot data were quantified from three independent experiments.

### smFISH

Cells were fixed with 4% (w/v) paraformaldehyde (Supplementary Data 1) for 30 min at RT. After fixation, cells were washed three times with PBS, and free aldehyde groups were quenched with 0.1% (w/v) $NaBH_4$ in PBS for 10 min at RT. Cells were then washed three times with PBS and further quenched with 100 mM glycine in PBS for 10 min at RT and washed three times with PBS. Next, cells were permeabilized with 0.2% (v/v) Triton X-100 (Supplementary Data 1) for 15 min at RT followed by five washes with PBS. smFISH was performed using ViewRNA high-content screening assay and signal amplification kits (Supplementary Data 1) as previously described[70] and according to manufacturer's instructions with some modifications. Specifically, protease treatment was not performed, as smFISH was followed by immunofluorescence or iterative indirect immunofluorescence imaging (4i). After the permeabilization, cells were incubated with the gene-specific probe sets (Type 1 and 6) for 3 h at 40 °C, washed three times with the wash buffer, incubated with the PreAmp probes for 1 h at 40 °C, washed as above, incubated with the Amp probes for 1 h at 40 °C, washed as above, incubated with the Label probes for 1 h at 40 °C, and washed as above. Nuclear DNA was stained using 4′,6-diamidino-2-phenylindole, dihydrochloride (DAPI) (Supplementary Data 1) for 10 min at RT at a final concentration of 0.4 µg/mL in PBS. Then, cells were washed three times with PBS and stored in PBS with azide (Supplementary Data 1). All aspiration and dispensing steps were performed using EL406 BioTek washer-dispenser (Agilent), and the aspiration was done to 30 µL.

After the imaging, smFISH was followed by immunofluorescence or 4i, and thus the smFISH signal was removed using the elution buffer (0.5 M L-glycine, 3 M urea, 3 M guanidinum chloride, 70 mM TCEP-HCl, pH 2.5) that originates from the 4i protocol[16]. Cells were first washed three times with PBS and then incubated in the elution buffer for 30 min at RT. Elution buffer was changed every 10 min. Efficient removal of the signal was verified by imaging. After this, cells were washed four times with PBS and proceeded to the blocking step of the immunofluorescence or 4i protocol.

### Immunofluorescence

For immunofluorescence imaging, cells were fixed as described above for smFISH and quenched with 500 mM $NH_4Cl$ in PBS for 10 min at RT. Permeabilization was done as for smFISH and was followed by blocking in 3% (w/v) BSA in PBS for 1 h at RT. Cells were then incubated with the primary antibodies for 2 h at RT followed by an incubation with the secondary antibodies for 1 h at RT (Supplementary Data 1). Antibodies were diluted in 3% (w/v) BSA in PBS. Nuclear DNA was stained with NucBlue Fixed Cell ReadyProbes Reagent (DAPI) (Supplementary Data 1) in PBS (4 drops/mL) for 10 min at RT, and total protein was stained with Alexa Fluor 647 NHS Ester (succinimidyl ester) (Supplementary Data 1) for 10 min at a final concentration of 0.2 µg/mL in 50 mM carbonate-bicarbonate buffer (pH 9.2). Cells were washed three times with PBS between each step, except that after the antibody incubations four washes were performed. Samples were stored in PBS with azide. If cells were stained with anti-p-STAT1 or anti-PCNA antibodies, a second permeabilization step using 0.1% (w/v) SDS in PBS for 10 min at RT was performed after the Triton X-100 step.

4i was performed after the smFISH and was done as previously described[16] with some modifications. In every 4i cycle cells were (1) blocked, (2) stained with primary and secondary antibodies as well as with DAPI, (3) imaged, and then (4) the signal was eluted. (1) Cells were washed four times with PBS and blocked in Intercept blocking buffer (Supplementary Data 1) supplemented with 100 mM $NH_4Cl$, 150 mM Maleimide, and 5% (v/v) donkey serum (Supplementary Data 1) for 1 h at RT. (2) Cells were washed three times with PBS and incubated with the primary antibodies for 2 h at RT. Cells were then washed four times with PBS and incubated with the secondary antibodies for 1 h at RT.

Antibodies were diluted in Intercept blocking buffer supplemented with 100 mM $NH_4Cl$. Cells were subsequently washed four times with PBS, and nuclear DNA was stained as in smFISH. Then, cells were washed four time with PBS and, (3) imaged after adding the imaging buffer (700 mM N-Acetyl-Cysteine, 200 mM HEPES pH 7.4). (4) Antibodies were eluted using the elution buffer as described for smFISH. In the last cycle cells were stained after the DAPI staining with the total protein staining (succinimidyl ester) as described for immunofluorescence above.

## Microscopy

Samples from the smFISH and 4i experiments were imaged on an automated spinning-disk confocal microscope (Yokogawa CellVoyager 7000) using a 40×/NA0.95 air objective, four excitation lasers (405, 488, 568, and 647 nm), and two Neo sCMOS cameras (Andor). Per site 12 confocal Z-slices with a 1-μm z-spacing were acquired, and all images were maximum projected during acquisition.

For other experiments images were acquired on a confocal microscope from Olympus (IXplore SpinSR10) with a Yokogawa spinning disk using a 40×/NA0.95 air objective, two excitation lasers (405/ 561 and 488/640 nm), and two ORCA-Fusion sCMOS cameras (Hamamatsu), or on an automated spinning-disk confocal microscope from Molecular Devices (ImageXpress Confocal HT.ai) using a 20×/NA0.95 or 40×/NA1.15 water objective, four excitation lasers (405, 470, 555, and 638 nm), and a CMOS camera. Per site 10–18 confocal Z-slices with a 1-μm z-spacing were acquired. On the ImageXpress device, images were maximum projected during acquisition. For images acquired on Olympus spinning disk microscope, maximum intensity projection was computed using Olympus cellSens (Supplementary Data 1), and two-channel images were split to one-channel images using Fiji. Imaging on the Olympus spinning disk and ImageXpress was performed with equipment maintained by the Center for Microscopy and Image Analysis, University of Zurich.

Microscopy images shown in figures were created using napari (Supplementary Data 1).

## Segmentation and feature extraction

TissueMaps (Supplementary Data 1), an open-source project for high-throughput image analysis developed in the Pelkmans laboratory (University of Zurich), was used for microscopy image preprocessing, object segmentation, and single-cell feature extraction. First, microscopy images were corrected for illumination artifacts as previously described[72]. If images were acquired in multiple cycles, they were aligned using DAPI, or DAPI and H2B signal, relative to the first cycle as previously described[16]. Nuclei were then segmented by (1) the Otsu thresholding of DAPI signal, (2) filling holes, (3) separating clumps, and (4) removing small objects. Labeled nuclei were used as seeds in the segmentation of cells. First, total protein signal from succinimidyl-ester staining was smoothed using mean filter. Cell outlines were then detected using the watershed transform of the smoothed intensity and adaptive thresholding[72]. For BJ cells, only nuclei were segmented.

For the smFISH experiment of HSV-1 and cellular transcripts and smFISH + 4i experiment, a pixel classifier was trained in Ilastik (Supplementary Data 1) to aid in the separation of nuclei clumps. The classifier used CTNNB1 signal to detect cell outlines. The resulting pixel probability maps were then uploaded in TissueMaps and smoothed using bilateral filter. DAPI signal was smoothed using the same filter and masked with the inverted cell outlines obtained from the pixel classification. Nuclei were segmented as described above without the clump-separation step. To capture the whole nucleus after masking the DAPI signal with the cell outlines, the resulting objects were first shrinked and then used as seeds to segment complete nuclei by propagation method and smoothed DAPI signal. Total protein and calreticulin signals were smoothed using gaussian filter and then combined.

The resulting new intensity image was masked with the inverted cell outlines obtained from the CTNNB1 pixel classification, and cells were segmented using nuclei as seeds and the watershed transform of the masked intensity images as above. To capture the whole cells, the resulting objects were used as seeds and the segmentation was repeated with the unmasked combination of total protein and calreticulin signal. In the smFISH + 4i experiment, α-tubulin signal was smoothed by gaussian filter, and the unmasked combination of total protein, calreticulin, and α-tubulin was used in this second segmentation step of the cell. Cytoplasm was segmented by masking cell segmentation with nucleus segmentation.

PML and P bodies were segmented using the Otsu thresholding of SP100 or DDX6 signal, respectively, and by separating clumps and removing small objects. For PML bodies objects outside the nucleus were excluded. To segment splicing speckles and nucleoli, a pixel classifier using SRSF2 or C23 signal, respectively, was trained in Ilastik, the resulting probability density maps were smoothed in TissueMaps using gaussian filter, manually thresholded, and then small objects as well as objects outside the nucleus were excluded. The same approach was followed to segment the viral RCs using ICP8 signal or the RCs were segmented without the pixel classifier using the Otsu thresholding of ICP8 signal and by removing small objects and objects outside the nucleus. Thus, in this study the segmented RCs represent ICP8-positive late RCs as dot-like ICP8-positive structures (early RCs) were not segmented.

TissueMaps was used to extract intensity, texture, and morphology features from segmented cells, nuclei, cytoplasm, and cellular and viral compartments. In addition, neighbor features were measured.

Computational detection of smFISH spots in TissueMaps was performed as previously described[72]. Deblending was used to separate adjacent spots. The same approach was used to detect virions in the ICP5-stained cells.

Support-vector machines (SVMs) were used to classify cells in TissueMaps and followed the same principles as CellClassifier[73]. First, cells representing two different classes were manually selected, and supervised machine learning models were trained using these example cells and a subset of single-cell features. Classification result was visualized, new example cells were selected, and the process was repeated until most cells were correctly classified. Further classification was performed in R (Supplementary Data 1).

## Data clean-up and normalization

Border cells, i.e. cells touching image borders, and missegmented cells, i.e. cells with unsegmented nucleus in the cytoplasm, were removed from all datasets. Missegmented cells were classified in TissueMaps using SVMs. TissueMaps also labeled border cells.

smFISH spots in the nucleus, cytoplasm, and cell area reflect the spots that overlapped with the corresponding segmentations. However, it is important to note that the smFISH method used in this study detects only cytoplasmic transcripts due to nuclear inaccessibility of the branched-DNA probes[70]. Thus, the transcript count in the cell area is used as a cytoplasmic transcript count. Point pattern of segmented smFISH spots within a cell was measured in TissueMaps and mean of relative distance to the cell border was used to divide cells to two groups: (1) cell was considered as expressing the gene detected by smFISH if distance was at least 0.01, or (2) cell was labeled as containing smFISH spots from a neighbor due to missegmentation of the cell border if the mean of relative distance was smaller than 0.01. Only cells classified in group 1 and having at least 5 spots were included in the analyses of *UL29* and *UL19* spot counts in Figs. 1 and 2 and Supplementary 1.

The smFISH + 4i experiment contained five time points, two mock-infected and four HSV-1-infected wells per time point, two mRNAs, 34 antibodies, DAPI and succinimidyl ester stainings. Furthermore, two additional mock wells per time point were stained with

secondary but no primary antibodies. Single-cell mean intensity measurements showed slight variation between replicates, most likely due to technical variation of the automated liquid handling system, and this was corrected by multiplying intensity and texture-feature values by a well-specific correction factor. First, median value of a feature was extracted per well, and then mean of the median values from all replicate wells was calculated. Correction factor was obtained by dividing the replicate mean value by the well median value. The same correction was performed in the smFISH experiment of HSV-1 and cellular transcripts for the intensity values.

Background was subtracted from the intensity values. For cellular markers, background value was either (1) from well regions without cells or (2) from cells stained with secondary but no primary antibodies. The first approach was used for DAPI and succinimidyl ester stainings and for antibody stainings when no multiplexing was performed. The second approach was used for all cellular 4i markers. For viral markers, background value was from mock cells stained with corresponding antibodies.

In the smFISH + 4i experiment one plate represented one time point and some plate-to-plate variation was detected in the feature values. This variation mostly comes from the imaging buffer that reduces signal intensity of some antibodies the longer they are in the buffer, and imaging of all five plates typically took ~12–15 h. The imaging sequence in all cycles was from 12-hpi plate to 1.5-hpi plate. Thus, it was important to correct this plate-to-plate variation for the analyses in which cells from multiple time points were combined. First, outlier cells with extremely high intensity values were removed by excluding cells that had a mean intensity higher than the 99.995th percentile. All cellular features were then corrected for the plate-to-plate variation by standardizing relative to the mock cells at each time point: z-scoring by subtracting the mean and dividing by the standard deviation derived from the corresponding mock cells. This should also remove changes that were not HSV-1-induced. All cellular features were further normalized relative to the uninfected cells without the infected neighbors within the same plate by subtracting their mean and dividing by their standard deviation within a time point. This corrected for staining differences between wells. The viral features were normalized by standardizing relative to all cells from the HSV-1-infected wells.

In the smFISH experiment of HSV-1 and cellular transcripts the cellular features were normalized by standardizing relative to mock cells.

In the immunofluorescence experiments the cellular features were normalized by first standardizing relative to mock cells and then relative to the uninfected cells without the infected neighbors. For BJ cells, the second normalization step was performed relative to all uninfected cells. The viral features were normalized as described for the smFISH + 4i experiment.

Log$_2$-transformed values shown in figures were not standardized.

## Cell classification

Cell-cycle classification was performed as previously described[16,74]. Cells were first classified as mitotic or non-mitotic using SVMs trained on intensity and texture features of nuclear DAPI and morphology features of nucleus and cell. S-phase cells were identified using SVMs trained on intensity and texture features of nuclear DAPI and PCNA. Subsequently mitotic and S-phase cells were excluded, and the remaining cells were classified as G1 and G2 using K-means clustering and total nuclear intensity of DAPI from the first imaging cycle. K-means clustering was done in R using function *kmeans* (package *stats*). Mitotic cells were removed from all datasets. If PCNA staining was not performed, cells were classified into G1, G1/S, G1/S/G2, S/G2, and G2 using the normalized total nuclear intensity of DAPI and 25th, 55th, 70th, 85th percentiles.

To classify apoptotic cells an SVM was trained using mean intensity of cleaved caspase 3 signal and texture features of total protein staining.

In the smFISH + 4i experiment cells were assigned infected using two approaches depending on the time point: (1) at 1.5 hpi, a cell was labeled infected if it contained at least one ICP5-positive spot, and (2) at 3–12 hpi, a cell was classified infected if it expressed one or more of ICP0, ICP4, or ICP27. In the immunofluorescence experiments cells were classified as infected if they expressed ICP27. Cells expressing ICP27 were classified using SVMs that were trained on mean and minimum nuclear intensity of ICP27 signal. Cells expressing ICP0 were identified using SVMs trained on mean and minimum intensity and texture features of ICP0 signal. Cells expressing ICP4 were identified using SVMs trained on mean and minimum nuclear intensity and nuclear texture features of ICP4 signal. Cells expressing ICP8 were identified using SVMs trained on mean and minimum nuclear intensity of ICP8 signal.

Besides classifying cells to uninfected and infected, uninfected cells were further divided into two subpopulations based on their location in the cell population: (1) uninfected cells without infected neighbors and (2) uninfected cells with infected neighbors. First, a neighbor infection score was calculated by extracting all neighbors for one cell determined in TissueMaps and then dividing the number of infected neighbors by the number of all neighboring cells. If a cell was uninfected and its neighbor infection score was larger than zero, the cell was classified in subpopulation 2. Otherwise, uninfected cells were classified in subpopulation 1. The classification was performed in R.

To classify infected cells to different infection stages, K-means clustering of cells that contained no viral replication compartments was performed using normalized mean intensity of ICP4 and ICP27. Optimal number of clusters (k) was determined by computing total within-cluster sum of square and plotting these values versus the number of clusters. k = 4 resulted in the bend in the elbow ("elbow method") and was used in the clustering. K-means clustering was done in R using function *kmeans* (package *stats*).

## Statistics and reproducibility

smFISH experiment of HSV-1 and cellular transcripts was performed using one biological replicate with five technical replicates, and smFISH + 4i experiment was performed using one biological replicate with two (mock infection) or four (HSV-1 infection) technical replicates. Immunofluorescence experiments were performed using 1-2 biological replicates with 1–3 technical replicates. Western blot experiments were performed using 3 biological replicates. No statistical method was used to predetermine sample size. Number of replicates was chosen based on previous experiments carried out in our laboratories[16,25,75]. In image-based experiments, cells at image borders, missegmented cells and mitotic cells were removed from datasets. In addition, cells with extremely high intensity values were removed by excluding cells that had a mean intensity higher than the 99.995th percentile.

Microscopy images from the smFISH + 4i experiment shown in figures are representative images from four technical replicates. Microscopy images from IF experiments shown in figures are representative from at least two biological replicates.

Quantitative data analysis of the single-cell features extracted from TissueMaps was performed in R.

Coefficient of variation (CV) of single-cell transcript counts was calculated as the standard deviation of dataset divided by the mean of dataset.

Statistical significance of the fold change in the Western blotting was analyzed using one-way ANOVA with Tukey's multiple comparison post-test, and tests were performed in R using functions *aov* and *TukeyHSD* (from package *stats*).

Overlap of two probability distributions was computed using R function *overlap* (package *overlapping*). For statistical tests, normal distribution of variables was tested using R function *shapiro.test* (package *stats*) and equal variance using R function *var.test* (package *stats*). Marker distributions were compared between cell subpopulations using unpaired, two-sided Mann–Whitney U test (R function *wilcox.test* from package *stats*) or pairwise two-sided Kolmogorov–Smirnov (KS) test (R function *ks.test* from package *stats*). For KS test, all subpopulations were first randomly subsampled to 1000 cells, and then KS test was performed with these subsets. This was repeated five times, and the mean p value of five tests is indicated in the figures.

Statistical significance of the difference in the fraction of cells in different cell-cycle phases was analyzed by unpaired two-sample t-test and performed in R using function *t.test* (package *stats*) and "two-sided" as alternative hypothesis and Welch's correction if there was no equality of variances. In Fig. 2c, d, one well at 9 hpi was omitted from the cell-cycle classification as the PCNA intensity was lower compared to other wells resulting in an incomplete detection of S-phase cells.

Statistical significance of the difference in the percentage of apoptotic cells was analyzed by unpaired two-sample t-test and performed in R using function *t.test* (package *stats*) and "greater" as alternative hypothesis and Welch's correction if there was no equality of variances.

Number of cells expressing ICP27 after DMSO or MEKi treatment was normalized to combine data from two experiments. First, cell counts per well were derived, and then the mean of these counts per treatment and time point was calculated. Then, the mean from two experiments was derived. A correction factor per experiment was calculated by dividing the mean from two experiments by the mean of the corresponding experiment. All cell counts were then multiplied by these correction factors. Statistical significance of the difference in the percentage of ICP27-expressing cells was analyzed by unpaired two-sample t-test and performed in R using function *t.test* (package *stats*) and "two.sided" as alternative hypothesis.

Statistical information of the tests is reported in Supplementary Data 2.

## Binomial logistic regression

In the smFISH + 4i experiment cells were classified infected based on the presence of ICP5-positive spots (at 1.5 hpi) or on the expression of HSV-1 immediate early proteins (at 3–12 hpi). Binomial logistic regression with different combinations of single-cell features as independent variables was then used to predict whether a cell is infected or uninfected. Only cells exposed to HSV-1 were included in the analysis, and all viral features were excluded from the predictors. At each time point, equal numbers of uninfected and HSV-1-infected HeLa cells were randomly subsampled and combined (1.5 hpi, $n = 17,330$; 3 hpi, $n = 5488$; 6 hpi, $n = 15,074$; 9 hpi, $n = 18,764$; and 12 hpi, $n = 19,828$) resulting in a total cell count of ~80,000. For each time point a binomial logistic regression model was trained on three replicates and used to predict in the fourth replicate whether a cell is infected. Predictions were repeated for each replicate and mean of accuracy of four predictions is shown in Fig. 1e. All predictors were first normalized as explained in '*Data clean-up and normalization*', and regression analysis was then performed using R function *glm* (package *stats*). Accuracy of predictions was calculated as (true positive + true negative)/(true positive + false positive + true negative + false negative). When all cellular features from the smFISH + 4i experiment (3136 features) were used as predictors, principal component analysis (PCA) was first performed using R function *prcomp* (*stats* package) and principal components (PCs) explaining 95% of variance in the dataset were used as predictors. To select individual cellular features, which could predict cell fate, predictive power score (pps) using R package *ppsr* and a decision-tree regression model was determined between uninfected/

infected cell type and each of 3136 single-cell features extracted from the smFISH + 4i experiment. Seven single-cell features in Fig. 1e represent correlators with pps ≥ 0.3. These were used as single independent variables in the logistic regression and are the following Haralick texture features: DDX6, difference entropy (Kernel size: 2) in cell; p-CDK9, entropy (Kernel size: 10) in nucleus; RNAP II S2P, information measures of correlation (Kernel size: 2) in nucleus; RNAP II S5P, sum entropy (Kernel size: 2) in nucleus; total protein staining, correlation (Kernel size: 10) in nucleus; NRF2, angular-second moment (Kernel size: 2) in nucleus; and SP100, sum variance (Kernel size: 2) in nucleus. In *Cell size, cycle, context*, the following single-cell features were used as independent variables: cell cycle, cell size, position at a cell-islet edge, local cell density, and fraction touching neighbors.

## Pearson and Spearman correlation

Pairwise correlation of total intensities of viral markers (ICP0, ICP4, ICP27, ICP8, ICP5, and VP16) and viral mRNA spot counts per cell (*UL19* and *UL29*) with 3136 single-cell features extracted from the smFISH + 4i experiment was calculated using Pearson correlation. Viral marker intensities and cellular features were first normalized as explained in '*Data clean-up and normalization*', and then the Pearson correlation was computed using R function *rcorr* from package *Hmisc*. The cellular features that gave Pearson correlation coefficient (r) ≥0.3 or ≤−0.3 were selected (Supplementary Data 3), and the Spearman correlation of viral features with these was computed using R function *rcorr*.

## UMAP analysis of the smFISH + 4i data

Normalized single-cell features were first divided into two categories (Supplementary Fig. 2c, d): (1) cellular features ($n = 3136$) and (2) viral features ($n = 750$). Cellular features included intensity and texture features from 28 cellular markers as well as from DAPI and total protein staining, morphology features of cell, nucleus, and cytoplasm, count of subcellular compartments (splicing speckles, nucleoli, PML bodies, and PBs), neighborhood features, and cell-cycle information. Viral features included intensity and texture features from eight viral markers, RC count, *UL29* and *UL19* spot counts, and neighbor infection score. PCA was performed on both feature sets using R function *prcomp*, and PCs explaining 95% of variance in each dataset were used in UMAP analysis that was performed using R package *umap* with parameters n_neighbors = 15 and min_dist = 0.05.

Movies of UMAPs were generated using R package *gganimate*.

## Linear and multiple linear regression

In the smFISH + 4i experiment, linear regression (LR) and multiple linear regression (MLR) were performed with viral gene expression as a response variable and cellular features as predictors. Prediction was performed in cells classified infected (see above '*Cell classification*'). Both viral gene expression and cellular features were first normalized as described in '*Data clean-up and normalization*' except that *UL29* and *UL19* spot counts were not standardized.

In MLR, four combinations of predictors were used: (1) simple phenotypic state composed of 360 single-cell features that included intensity and texture features from DAPI and total protein stainings, area and morphology features of cell, nucleus, and cytoplasm as well as neighborhood and cell-cycle features, (2) 495 intensity and texture features of top Pearson-correlator markers (p-Akt, DDX6, HCFC1, RNAP II, and RNAP II S5P), (3) 495 intensity and texture features of five randomly chosen cellular markers (EEA1, NRF2, p-ERK, CRT, and GM130), and (4) high-dimensional cellular state space composed of 3136 single-cell features as described in the UMAP analysis. First, a dimensionality reduction using PCA (R function *prcomp*) was performed and then PCs explaining 95% of variance in each dataset were used in the MLR analysis.

LR and MLR were performed per time point for 3–12 hpi and for pooled time points combining cells from 1.5–12 hpi. Per time point,

regression models were trained using three replicates (R function *lm* from *stats* package) and prediction was performed on the fourth replicate (R function *predict* from *stats* package). Per pooled time points, LR and MLR models were trained on 15 replicates and prediction was performed on five remaining replicates so that both training and prediction data sets contained three or one replicate per time point, respectively. Model performance was measured by $R^2$ that represents the square of correlation between the predicted and measured values (R function *R2* from *caret* package). This was repeated for each replicate, and $R^2$ shown in Fig. 3g and Supplementary Fig. 7b–d is a mean of four predictions.

In the smFISH experiment of HSV-1 and cellular transcripts, MLR was used to predict viral or cellular gene expression using the simple phenotypic state as a predictor. MLR was performed as described above, except that 331 single-cell features describing the cell phenotypic state were used and total number of replicates was five. Each replicate was first subsampled to 2500 cells to allow comparison of viral and cellular transcription. $R^2$ shown in Supplementary Fig. 1c is a mean of five predictions.

To fit robust linear regression models to the single-cell data, R function *lmrob* from *robustbase* package was used.

### MCU analysis

In the smFISH + 4i experiment 5500 cells per time point were randomly selected from the HSV-1-infected wells, and construction of MCUs for these 27,500 cells was performed as previously described[16] with some modifications. Multiplexed pixel profiles (MPPs) were extracted from TissueMaps after the illumination correction, alignment, and segmentation using Python scripts mpp.py and pixel_profiles.py (Supplementary Data 1). MPP contained the intensity values for 26 cellular markers, DAPI, and total protein staining (Supplementary Fig. 2b). Calreticulin and TGN46 were excluded due to the nuclear background. In order to correct intensity differences between plates (i.e. time points, see above 'Data clean-up and normalization') MPPs were multiplied by a time-point specific correction factor. First, uninfected cells without infected neighbors were selected and median of the single-cell mean intensity was calculated per time point. Correction factors were then derived by dividing the average median of all time points by the median per time point. MPP files contained ~900 million pixels. Next step was clustering of MPPs using self-organizing maps (SOMs). For this step MPP files were randomly subsampled to 200 cells per time point resulting in a total of ~36 million pixels. These were used to train a SOM model that was then applied to all pixels. SOM clustering was performed using Python script combine_mpp_and_cluster.py (Supplementary Data 1). The final step was to cluster SOM nodes to MCUs using R (*igraph* and *leiden* packages and the Leiden clustering algorithm) with four nearest neighbors, which resulted in the identification of 44 MCUs.

Marker intensities in MCUs were calculated by first z-scoring intensities per marker and MCU (R function *scale*) and then calculating mean intensity per MCU. To cluster markers based on their intensity in each MCU, a distance matrix was first computed using Euclidean distance matrix (R function *dist* from package *stats*) followed by hierarchical clustering (R function *seriate* from package *seriation*). To cluster MCUs based on their marker intensity profiles the same hierarchical clustering was performed.

MCU abundance (i.e. fraction of pixels that belong to each MCU of total pixels in a cell) in each cell subpopulation per time point was computed by grouping pixels from each cell subpopulation per time point and dividing the number of pixels in each MCU by the total number of pixels in that subpopulation. This is represented as a stacked bar plot in Fig. 4b.

Pairwise spatial proximity scores (SPSs) between all MCUs were calculated per cell as described previously[16] using codes available at https://doi.org/10.17632/ytvttnr2nn.1. For each MCU, neighboring pixels assigned by 8-connectivity were grouped based on their MCU identity and the number of neighboring pixels in each MCU group was divided by the total number of neighbors. To assess significance of the SPSs in the original cells, we randomly permutated the original MCU identities of pixels, not the coordinates, per cell and then recalculated a randomized SPS. The permutations and calculations were repeated 1000 times per cell. The mean of these randomization controls was then subtracted from the original SPS per MCU-interaction pair per cell. Next, mean of the normalized SPS values was computed per MCU-interaction pair per all cells, and MCUs were clustered using the mean SPSs and UMAP analysis that was performed using R package *uwot* with parameters n_neighbors = 4, metric = "manhattan", spread = 1.8, and min_dist = 0.1. In Fig. 4c mean SPS per MCU was computed using 27,500 cells (uninfected and infected cells from all time points) and in Fig. 4d using 1520 cells (infected cells from 12 hpi). The size of MCUs presented in UMAPs (Fig. 4c, d) was calculated by collecting pixels from the corresponding cells, grouping them per MCU, and then normalizing the number of pixels in each MCU to the largest MCU. Coloring of MCUs in the UMAPs represents their abundance in the indicated groups of cells. First, abundance of MCUs was computed per cell by dividing the number of pixels of each MCU by the total number of pixels in the corresponding cell, and then mean of the abundances per indicated group of cells was calculated.

Infected cells at 12 hpi (1520 cells) were divided into subpopulations using MCU abundance and Leiden clustering (R packages *igraph* and *leiden*). Eight nearest neighbors resulted in the identification of 13 subpopulations. Euclidean distance matrix and hierarchical clustering were performed in R using *pheatmap* package. For MCU abundance and viral gene expression presented in Fig. 4e heatmaps, median of each value within a subpopulation was first computed and then features were z-scored using the mean and standard deviation of all subpopulations (R function *scale*). Standardized values were then min-max-normalized using R functions *preProcess* (package *caret*) and *predict* (package *stats*).

### Computational infrastructure

Image analysis was performed on the high-performance clusters (ScienceCloud and ScienceCluster) provided by the Service and Support for Science IT (S3IT) facility of University of Zurich. SPS calculations were performed with the computational resources provided by the ELIXIR node, hosted at the CSC–IT Center for Science, Finland.

### Reporting summary

Further information on research design is available in the Nature Portfolio Reporting Summary linked to this article.

## Data availability

Source data are provided with this paper (Supplementary Data 1). Western blot, single-cell, and single-pixel datasets generated during the current study have been deposited at Mendeley, and the DOIs are listed in Supplementary Data 1. Raw microscopy image datasets used in this study contain also information not reported and will be shared upon reasonable request. Any additional information required to reanalyze the data reported in this paper is available upon request.

## Code availability

All original code has been deposited at Mendeley, and the DOI is listed in Supplementary Data 1. Codes for image analysis performed using TissueMAPS are found at https://github.com/pelkmanslab/TissueMAPS. Python-based code for calculating population-context features and multiplexed cell units (MCUs) was written on previous MATLAB code[16,25,76] and is available at https://github.com/scottberry/popcon and https://github.com/scottberry/mcu, respectively (Supplementary Data 1). R codes for the analyses of single-cell features and single-pixel intensities were developed on a per-experiment and per-figure basis and are available on request.

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

## Acknowledgements

We thank Scott Berry for providing codes for and help with MCU analysis, Bernd Vogt for his help in HSV-1 purification, and all members of Cornel Fraefel's and Lucas Pelkmans' groups for discussions. A549 cells were kindly provided by Benjamin G. Hale. M.K.P. was supported by Sigrid Jusélius Fellowship grant. L.P. was supported by the Swiss National Science Foundation grant 310030_192622, the European Research Council advanced grant CROSSINGSCALES-885579, the Chan Zuckerberg Initiative grant CZF2019-002440, and the University of Zurich. C.F. was supported by the Swiss National Science Foundation grants 310030_184766 and 310030_212248.

## Author contributions

Conceptualization, M.K.P., L.P., and C.F.; methodology, M.K.P.; investigation, M.K.P., J.B., and J.R.; formal analysis, M.K.P., J.B., and J.R.; resources, L.P. and C.F.; visualization, M.K.P.; writing – original draft, M.K.P.; writing – review and editing, M.K.P., J.R., L.P., and C.F; funding acquisition, M.K.P., J.R., L.P., and C.F.

## Competing interests

L.P. has filed a patent on the 4i technology (patent WO2019207004A1), consults for Dewpoint Therapeutics, and has ownership interest in Sagimet Biosciences and Apricot Therapeutics. The remaining authors declare no competing interests.
