## [Peer Review File · Nature Communications]

Cellular state landscape and herpes simplex virus type 1 infection progression are connectedReviewer #1 (Remarks to the Author):

Dear editorial team at Nature Communications and authors,

Thank you for the kind invitation to review the manuscript "Cellular state landscape predicts virus infection progression". In this work, Pietilä and colleagues delivered an interesting article about outcome heterogeneity upon viral infection. In brief, this study addresses the heterogeneity of HSV-1 infection outcomes using high-dimensional RNA and protein mappings to characterise cellular hallmarks that the hosts display in response to infection. To achieve this, multiplexed imaging and quantitative single-cell measurements were employed across a time course following viral infection.

The overall work is convincing and relevant, but major comments arise regarding the choice of the cell type used for the study, the experimental design and the statistics used. For this reason, and although positive about the work, the manuscript requires significant revisions, and the authors required to clarify specific points before it is accepted:

Major comments:

HeLa cells are probably not the best model to study the heterogeneity of viral infections since the cell line is extremely heterogeneous on its own (see <https://doi.org/10.1038/s41592-019-0375-1>). While we acknowledge the authors also tested a different cell line, using even more cell lines (particularly those whose cell type corresponds to natural targets of HSV-1 infection) would be beneficial to improve the robustness of the study. If this is not possible, the authors should clearly state the limitations of their study in the discussion.

Page 8, line 137: HSV-1 infections reduced the mean nuclear intensity of PCNA at later stages. This generally could result in variations of the cell cycle across infection such as arrest at the end of the G1 phase. Were individual cell cycle stages assessed to detect cell cycle arrest? If so, could this impact subsequent infections?

There is a lack of sample quantity throughout the manuscript. For example, Figure 1 and its data correspond to a single biological replicate, which, I would say, is only sufficient to quickly check if a phenomenon is worth pursuing, or to make statements along the lines of "this data suggests that". In other words, a single biological replicate is not sufficient to obtain statistically and biologically relevant results. In other words, there is not evidence that these results are reproducible. The authors carefully adapted their wording to not make claims that are not backed up by the data. This, however, means that the results might not have much biological relevance, simply because the authors chose not to repeat the experiment. Regarding the choice of wording, I suggest changing the word "implying" in line 95 to something like "suggesting".

The authors observed heterogeneity at the whole-population level and demonstrated that in their sample, ~80% of infections arise from a single virion, which was thought to minimise variation arising due to different virion numbers. However, when assessing heterogeneity in infection outcomes, it is important to actually exclude the cells infected by more than one viral particle from the analysis, or at least compare the number of cells infected by more than virus with the number of cells that actually deviate substantially from the average response

to infection, to demonstrate whether the heterogeneity happens in cells infected by one or more virions.

Page 8 line 126: The authors state that they needed to account for all features in the multivariate dataset to achieve high accuracy in predicting an infected cell. They state that this highlights the virus-induced multimodal change of cells. I think it only shows how conditional probabilities work: if a cell displays more than 1 trait associated with infection, then we can predict more accurately that it is infected if we take all the valuable traits into account. Indeed, the results show a multimodal change, but that is expected unless, as it is unlikely that a one viral infection would only impact a single cellular aspect.

In figure 2, the authors perform statistical tests to compare the means of groups of technical replicates, but they avoid this approach in Figure 1, despite the dataset being the same.

Figure 2b: The statistics shown are somewhat intriguing. How come groups “1.5 hpi” and “6 hpi” are different with statistical significance, while the “3 hpi” is not significant?.

Figure 2c: “The number of cells in G1, S, and G2 in each subpopulation was normalized to the total number of cells in each cell-cycle phase.” Do the authors mean “normalized to the total number of cells in that population”? I suggest adding a sentence to explain the normalization performed. The authors should clearly state what “less PCNA in the nucleus” suggests, for example, “less proliferation” or possibly “cell cycle arrest”. Again, it seems like no statistical tests were performed. The cells in S phase are clearly different in the last timepoint but it is unclear if that difference is statistically significant.

Figure 2d: The figure legend does not clearly mention the statistical tests performed. Why not perform some test showing non-significant results and actually state that instead of not doing statistical tests and just stating that “there was significant overlap”? Also, why multiply by 1000?

Line 146: One phenomenon hardly explains everything in most biological models. It is generally a combination of factors. This conclusion is somewhat irrelevant and is made throughout the paper.

Supplementary figure 4 is mentioned before supplementary figure 3.

When the authors perform linear regressions they choose to only report R^2 , which although the authors always correctly refer to it as “variance explained”, I believe the authors should add a sentence explaining why they based their conclusion on this single value.

Minor comments:

Figure 1a,b: The size and resolution of the image and the colours chosen (green, red) make it difficult to see the UL19 marker clearly.

Page 7, line 118: Has this taken into account the time the actual infection took place?

Page 7, line 122 - 124: Were there other markers? If so, how did the others fare?

Page 8, line 128: How was the cell cycle stage assessed?

Page 8, line 134: How is polymerase delta involved in the cell cycle? Stating this would be beneficial to the reader.

Page 8, line 137: There is still a lot of variability here that supports the idea of heterogeneous cell behavior following infection. However, it would be useful to know how many cells are involved in this measurement as the numbers are likely to change abruptly depending on them.

Page 9, line 157: Was the cell classification done manually? Please specify.

Page 9, line 158: How were the cell boundaries defined?

Page 11, line 221: Depletion is a strong statement that is not supported by the data. "rRduction" would better represent what's been shown.

Page 12, line 229: Why is there a large variability in cell numbers between infection stages?

Figure 7b: Labelling the two cell types on the figure itself could be beneficial to improve clarity.

Page 17, line 356-357: The wording is confusing

Reviewer #2 (Remarks to the Author):

In this manuscript, Pietilä and colleagues take a multiplexed imaging approach to investigate variability among cells infected by HSV-1. The work illuminates the high degree of variability during HSV-1 infection and, importantly, adds a spatial component to previously published reports by scRNAseq. I congratulate the authors on this beautiful work!

In addition to a few specific comments listed below, my main issue with the manuscript as it is currently written is the improper or not well-defined use of the term "prediction" and the improper conflation of correlation and causation. The authors extract information from cellular images and use multiple tools to correlate this information with the stage and extent of viral infection. While it is technically true that the authors use cellular features to predict the infection status of the cells, it is misleading and that this prediction does not suggest these cellular features are determinants of the infection status. Rather, these analyses show that different infection states are marked by different cellular states. Here are two examples:

Lines 208 - "multivariate source of infection heterogeneity" - While the authors clearly show that infection variability correlates to variability in the multivariate cellular space, they do not show that this is the source of infection heterogeneity.

Lines 421-422 - "we identified multiple cellular factors that explain single-cell variation in human herpesvirus infection". While these cellular factors explain variability in the data, this is a correlation and not a deterministic relation.

I highly recommend the authors rephrase their statements throughout the manuscript to prevent confusion between observed correlations and their underlying causes.

Specific comments:

1. Fig 1b, Fig 5 and Fig S4 - blurry, hard to read

2. Lines 114-119 : the authors describe that ~80% of infections originated from a single virion, and that a similar number of cells (~30%) was infected at 12 hrs and 1.5 hrs post-infection. I am confused by these numbers and would like to request the authors clarify the following points:

a. Since an MOI of 0.3 was used, how do the authors explain most cells have a single capsid in them? The ratio of genomes to PFU for HSV-1 is in the tens to hundreds, and the number of empty capsids are even higher. Wouldn't an MOI of 0.3 would translate to an average of >3 genomes per cells, and probably much more capsids?

b. My experience using live-imaging of HSV-1 infection suggests secondary infections are common from 6 hrs onwards. Could the authors explain why they don't see much secondary infections? Fig 1b also seems to suggest the number of infected cells increase with time.

3. From Line 235 - the section entitled "Multiplexed protein maps link cellular changes to infection heterogeneity" is unclear to me. I read this section several times, but could not understand the main points the authors were trying to make. I would suggest the authors clarify this section. However this might just be a failure on my part, so it's really up to the authors to decide if it needs clarifying.

4. Lines 356-358 - "Notably, HSV-1 induced the formation of PBs in neither cell lines (Fig. 7c,g), although this has previously been reported in HeLa cells". This sentence is unclear. Do the authors mean infection did not induce formation of PBs in both cell lines?

5. Lines 378-379 - the authors summarize that "HSV-1 infection progression depends on degradation of DDX6 and PBs", however this was not demonstrated in this study. While the authors have shown that during infection DDX6 and PBs are degraded, they did not show this is important for infection to progress.

In summary, this is an impressive and important work that further our understanding of the virus-host interactions at the single cell level, and I hope the authors find my comments useful.

Kind regards,
Nir Drayman

Reviewer #3 (Remarks to the Author):

Synopsis

Exploring and understanding heterogeneity in viral infections is key to increase our knowledge in this topic. The advent of high-throughput methods such as single-cell RNA-sequencing have greatly advanced this research, however the availability of spatial as well as protein/protein modification information is still sparse. In this study, the authors employ a multiplex immunofluorescence assay combined with a detailed, sub-cellular quantification of signal and signal patterns, in order to elucidate specific processes associated with Herpes simplex virus 1 infection in HeLa cells, such as ERK/Akt signaling and RNA polymerase II phosphorylation states. The data quality is very high, the ways the authors analyzed the data are excellent, and the biological findings of substantial interest. However, it is not clear with how many replicates the experiment have been performed, and to what extent the findings are reproducible, and also the data is often not presented adequately. Ideally, the findings should be reproduced in a second cell line such as the A549 cells employed in one experiment, or an independent experiment in HeLa cells be performed. Furthermore, clarity and level of detail of the data analysis should be improved as detailed below.

Major issues

- It is not clear to me how many replicates (i.e. different cell culture plate wells with separate processing) were done. Basically, important observations would need to be confirmed in independent experiments.
- Figure 1/S1: the signal for UL19 is clearly weaker than UL29, presumably because the copy number is lower. This could be either an intrinsic features of UL19 mRNA, or simply because the time points are too early to get the full UL19 signal. Is the low UL19 signal in S1 due dropouts, and therefore less reliable? Or could this be a sign of e. g. an abortive replication cycle? The authors should try to explain/discuss this observation.
- Along these lines, the raw data from the 4i experiment should be better characterized. This could e.g. include regression coefficients between all measured features as a supplementary table, with some (e.g. between viral proteins/RNAs, or everything that is above some threshold) highlighted in the
- From Fig. 2B, it looks as if PCNA protein levels are higher in infected cells at 1.5hpi. This would indicate that cells in G2 and S phase are more prone to infection, no?
- How many replicates (i.e. infected wells) make up the data in Figure 2? It would be important to see variability between independent experiments.
- Figure 3 is basically very nice and interesting, however using so many different projections make it difficult to understand. On the other side, there is very little quantitative information. For example, how many cells are in the various subpopulations? Which are the defining features of the island with the arrow in Fig. 3b? Is PML body presence associated with S phase (it looks a bit when comparing the leftmost with the rightmost plot in Fig. 3g, upper row)?
- Figure 4, there is an apparent discrepancy between Fig. 4a and 4c, in that in 4a the S5P signal is constantly going down over time (i.e. shifting to the left with progressing time) whereas in 4c the "late stage infection", which one would expect to be very prominent at 12 hpi, has again higher S5P signal. Furthermore, the cell populations are not well characterized. According to 4c, there are three populations (early/middle/late). Are these along a trajectory? What properties do these populations have, e.g. which values do the viral features in there show? Could there be more subpopulations than just these three? For example, there is quite a number of cells in 4e on the top right part of the UMAP (below the cells with lots of ICP8) with high S5P and S2P intensity, what is their status? Overall, the different cell populations need to be better characterized.
- Fig. 5a: Please explain the constructions of the MCU in a bit more detail so that readers to not have to go back to the original paper.
- Fig. 5: this figure contains the kind of data that I miss in Fig. 4. It is however not clear to me why cells are now classified differently compared to Fig. 4, as apparently the classification here is also along somehow along an infection trajectory. By far the best way in my opinion, in terms of clarity and traceability, would be to use the same classification of infected cells in Fig. 4 and 5, and describe the two different observations (Pol II phosphorylation, pathway activation) in the same subpopulations of cells.
- It would be most interesting to see how localized Pol II signal, particularly phosphorylation, looks

at viral replication centers. Can the authors add this information?

- Fig. 6 – again, the infected cell subpopulations are not well described. What are e.g. the properties of the cells with high p-Akt? Are there category 2 cells in 6c with high p-Akt (it looks a bit like an overlap of this triangular shaped part of the UMAP below the blue island).

Minor issues

- Abstract, the sentences/sentence parts “we mapped high-dimensional viral and cellular state spaces throughout the infection using multiplexed imaging and quantitative single-cell measurements.” and “The multiplexed protein maps of HSV-1-induced cellular modifications uncover” should be more specific. What did you image and measure?
- Introduction: Please try to rephrase the not-so-elegant first two sentences, and reconsider whether it is necessary to explicitly mention one disease (COVID-19).
- Introduction: “Infections are often asymptomatic, but disease outcome can vary from mild oral herpes to severe encephalitis.” – whereas this sentence is per se not wrong, the clinical features of the key aspects primary infection/latency/reactivation should be depicted more accurately.
- Fig. S2: fextures -> textures
- Fig. S3B: how well does this distribution fit the theoretically expected distribution of entering virions?
- Fig. 2BC, as a sanity check please show the uninfected cells from all time points next to each others with statistical tests, as there should be no difference there (except if there would be effects on neighbouring cells, but this does not seem to be the case).
- Fig. 5b and 6c, explain in the figure legends what 1/2/3 means.
- Fig. 5, “Both infected cells and cells with high ICP27 expression” – what is the difference then? Are there cells with high ICP27 that are not classified as infected?
- Fig. 5, different column orders in 5a/e and different row orders in 5e left/right make it difficult to read. Since the clustering is not of immediate importance, it would be better to not cluster rows/columns in 5e.
- “thus our multiplexed data reveal that HSV-1 was unable to shut down the host defense in these low-viral load cells.” infers a causality that cannot be derived from the data.
- The entry to the section “HSV-1 infection activates signaling responses also in neighboring cells” is not very elegant, please try to rephrase.
- Last sentence, “...and reveals potential targets for antivirals.” – this is a bit far-fetched, and for sure does not make the paper any better.

Cellular state landscape and virus infection progression are connected

(Previous title: Cellular state landscape predicts virus infection progression)

Maija K. Pietilä, Jana Bachmann, Janne Ravantti, Lucas Pelkmans, and Cornel Fraefel

Response to referees

We appreciate the valuable and constructive comments the reviewers made on our manuscript and feel that they have led to significant improvements of our work. We have validated our main observations in altogether three different cell lines, and the new results support the conclusions we presented in the original manuscript. We have also clarified all the other concerns raised by the reviewers. In addition, minor changes were done to improve the manuscript. To meet the editorial requests, the manuscript was also modified according to the formatting instructions and reporting summary. All changes are shown in the manuscript text file with track changes.

The figures were also reordered to improve the flow, especially regarding to how cells were classified into different subpopulations in different analyses.

- Original Fig. 4 → 7
- Original Fig. 5 → 4
- Original Fig. 6 → 5
- Original Fig. 7 → 6

Besides, five supplementary figures were added, and supplementary figures reordered according to the main figures.

- Original Supplementary Fig. 4 → 5
- Original Supplementary Fig. 5 → 6
- Original Supplementary Fig. 6 → 7 and 9

If not stated otherwise, figure numbers in our responses refer to those in the revised manuscript. High-resolution figure files can be downloaded from:

<https://figshare.com/s/7151f081e8538f5f7ac3>.

Before detailed point-by-point responses to the reviewers' comments, we would like to present the following response as the reviewers raised concerns regarding the number of biological replicates:

- smFISH data presented in the manuscript is from two biologically independent experiments.

- Due to the long time and costs required to perform experimental work, imaging and data analysis, the smFISH + 4i experiment was performed only once, but using two (mock infection) or four (HSV-1 infection) replicate wells resulting in a total of ~243,000 single cells. In addition, similar 4i studies have previously been published with one biological replicate (Gut, G., Herrmann, M. D. & Pelkmans, L. *Multiplexed protein maps link subcellular organization to cellular states*. Science 361, eaar7042 (2018) and Kramer, B. A., Sarabia Del Castillo, J. & Pelkmans, L. *Multimodal perception links cellular state to decision-making in single cells*. Science 377, 642-648 (2022)).
- To validate our results, main findings from the smFISH + 4i experiment performed in HeLa cells were reproduced in three cell lines, two epithelial, cancer cell lines (HeLa and A549), and in one primary, fibroblast cell line (BJ). All three cell lines are commonly used in HSV-1 research. We repeated PCNA staining and cell-cycle analysis, effect of cell size in infection heterogeneity, p-Akt and p-ERK stainings, RNAP II stainings, and DDX6 staining and P-body analysis in these three cell lines. All results were successfully reproduced in HeLa cells, and results from A549 and BJ cells also support our conclusions.
 - For fibroblasts, only nuclei were segmented, and nuclear intensities of markers were quantified as segmentation of the cell border was not possible in 2D due to the significant overlap of these cells.
 - We would also like to emphasize that classification of cells as infected is different between the 4i and conventional immunofluorescence experiments. As multiple viral markers were stained in the same cells in the smFISH + 4i experiment, we could use information from all of them to identify infected cells. In the conventional immunofluorescence experiments, cells are classified as infected based on ICP27 expression as it is one of the two first genes expressed in HSV-1-infected cells, and in our smFISH + 4i experiment we also noticed that the majority of the cells classified as infected at 3-12 hpi were expressing ICP27, making it a reliable marker for infected cells. We also used ICP4 and ICP8 as a marker for the infected cells in the conventional immunofluorescence experiments, and the marker used is always indicated in the figures and figure legends.
- In addition, inhibitor experiments using U0126 (inhibits ERK activation) and LY294002 (inhibits Akt activation) were repeated in HeLa cells.
- We have now added the number of individual replicate wells, information about biologically independent experiments, and cell counts in the figures and figure legends.

Below please find our responses to the points raised by each reviewer.

Reviewer #1

Dear editorial team at Nature Communications and authors,

Thank you for the kind invitation to review the manuscript "Cellular state landscape predicts virus infection progression". In this work, Pietilä and colleagues delivered an interesting article about outcome heterogeneity upon viral infection. In brief, this study addresses the heterogeneity of HSV-1 infection outcomes using high-dimensional RNA and protein mappings to characterise cellular hallmarks that the hosts display in response to infection. To achieve this, multiplexed imaging and quantitative single-cell measurements were employed across a time course following viral infection.

The overall work is convincing and relevant, but major comments arise regarding the choice of the cell type used for the study, the experimental design and the statistics used. For this reason, and although positive about the work, the manuscript requires significant revisions, and the authors required to clarify specific points before it is accepted:

Major comments:

HeLa cells are probably not the best model to study the heterogeneity of viral infections since the cell line is extremely heterogeneous on its own (see <https://doi.org/10.1038/s41592-019-0375-1>). While we acknowledge the authors also tested a different cell line, using even more cell lines (particularly those whose cell type corresponds to natural targets of HSV-1 infection) would be beneficial to improve the robustness of the study. If this is not possible, the authors should clearly state the limitations of their study in the discussion.

- We acknowledge that HeLa cell lines can be heterogenous between laboratories. However, we chose to use the HeLa Kyoto cell line because it was used in a previous study showing that cellular transcription can be predicted (Battich, N., Stoeger, T. & Pelkmans, L. *Control of Transcript Variability in Single Mammalian Cells*. Cell 163, 1596–1610 (2015)). In this study by Battich et al., they also used primary human keratinocytes, and results from HeLa and keratinocytes were similar, indicating that these HeLa cells are suitable for cell-to-cell variability studies.
- To validate our results, we also validated the main findings in two other cell lines, A549 and BJ cells, as described above.

Page 8, line 137: HSV-1 infections reduced the mean nuclear intensity of PCNA at later stages. This generally could result in variations of the cell cycle across infection such as arrest at the end of the G1 phase. Were individual cell cycle stages assessed to detect cell cycle arrest? If so, could this impact subsequent infections?

- It is a good point that decreased PCNA mean intensity most likely reflects HSV-1-induced changes in the cell cycle, and this was clarified in the results.
- Cell cycle stages were determined at each time point and are shown in Fig. 2c and Supplementary 4b for HeLa cells. Cell cycle stages were also quantified in A549 and BJ cells after HSV-1 infection (Supplementary Fig. 4d,e). Data show that infection resulted in a decrease in the number of S-phase cells and increase in the number of G1- and G2-phase cells. This agrees with previous reports of HSV-1 induced cell cycle arrest at G1/S and G2/M. The cell-cycle classification results are also in agreement with the observed PCNA mean intensity decrease. The results were similar in all three cell lines tested.

There is a lack of sample quantity throughout the manuscript. For example, Figure 1 and its data correspond to a single biological replicate, which, I would say, is only sufficient to quickly check if a phenomenon is worth pursuing, or to make statements along the lines of “this data suggests that”. In other words, a single biological replicate is not sufficient to obtain statistically and biologically relevant results. In other words, there is not evidence that these results are reproducible. The authors carefully adapted their wording to not make claims that are not backed up by the data. This, however, means that the results might not have much biological relevance, simply because the authors chose not to repeat the experiment. Regarding the choice of wording, I suggest changing the word “implying” in line 95 to something like “suggesting”.

- We have added sample quantities throughout the manuscript and also emphasized from which experiments the data are derived from.
- We understand that two biological replicates of the smFISH + 4i experiment would make this study even more convincing. However, due to the nature of the 4i technique as explained above, we are using one biological replicate with 2 or 4 technical replicates. In addition, data were acquired from hundreds of thousands of cells, and the main findings were validated in biologically independent experiments as described above.
- Data in Figure 1 is from two biologically independent experiments, and in both experiments cell counts were tens of thousands, from four or five technical replicates.
- In line 95 (original manuscript), the word “implying” was changed to “suggesting”. In addition, the result shown in Supplementary Fig. 1c (the low prediction power of viral gene expression, when a simple cell phenotype is used) is repeated in a biologically independent experiment shown in Fig. 3g. Furthermore, this conclusion is also backed-up by our preliminary experiments.

The authors observed heterogeneity at the whole-population level and demonstrated that in their sample, ~80% of infections arise from a single virion, which was thought to minimise variation arising due to different virion numbers. However, when assessing heterogeneity in

infection outcomes, it is important to actually exclude the cells infected by more than one viral particle from the analysis, or at least compare the number of cells infected by more than virus with the number of cells that actually deviate substantially from the average response to infection, to demonstrate whether the heterogeneity happens in cells infected by one or more virions.

- We appreciate that we cannot completely exclude variation arising from the number of infecting virus particles, and this was clarified in the results. However, as ~80% of infections are single-particle infections and ~20% two-particle infections, one should expect to see a similar pattern in the heterogeneity of HSV-1 gene expression (Fig. 1c and Supplementary 3f), and this we didn't observe; but the heterogeneity greatly exceeded the heterogeneity in the number of infecting virus particles and formed a continuum of different expression levels. Furthermore, significant heterogeneity in the viral gene expression was observed even at MOI 0.06 that should result in an even higher percentage of single-particle infections compared to MOI 0.3 used in the smFISH + 4i experiment (Fig. 1c), indicating that the heterogeneity does not derive from the number of infecting viral particles when low MOIs are used.
- As these experiments were performed with fixed cells, we unfortunately cannot follow which cells originate from single-particle and which from multiple-particle infections after the first time point, and thus it is not possible to exclude cells that originate from multiple-particle infections at the late time points. However, this would be an interesting further idea to be performed by live-cell imaging, but this requires significant technological development so that multiplexed live-cell imaging could be performed.

Page 8 line 126: The authors state that they needed to account for all features in the multivariate dataset to achieve high accuracy in predicting an infected cell. They state that this highlights the virus-induced multimodal change of cells. I think it only shows how conditional probabilities work: if a cell displays more than 1 trait associated with infection, then we can predict more accurately that it is infected if we take all the valuable traits into account. Indeed, the results show a multimodal change, but that is expected unless, as it is unlikely that a one viral infection would only impact a single cellular aspect.

- This point was clarified in the results. However, we also think that it is important to emphasize the multimodal change of cells to show that among the markers used here, there is no single feature that could predict cell fate as well as high-dimensional cellular state.

In figure 2, the authors perform statistical tests to compare the means of groups of technical replicates, but they avoid this approach in Figure 1, despite the dataset being the same.

- In Fig. 2b, different cell subpopulations (uninfected and infected cells) are compared and statistical tests were used to determine if differences between these subpopulations are statistically significant. In Fig. 1c, the goal is to show the

heterogeneity in HSV-1 *UL29* and *UL19* expression and no comparison is made between the two transcripts and thus no statistical tests were performed for Fig. 1c data.

Figure 2b: The statistics shown are somewhat intriguing. How come groups “1.5 hpi” and “6 hpi” are different with statistical significance, while the “3 hpi” is not significant?.

- This discrepancy is explained by how cells were classified as infected. At 1.5 hpi cells were classified as infected if they contained incoming virions while at 3-12 hpi cells were classified as infected if they expressed immediate early proteins. Thus, at 3 hpi infected cells represent only a fraction of those cells detected infected at 1.5 hpi, the fraction in which viral gene expression was detectable. At 6 hpi viral gene expression was detectable in a larger fraction that was similar to that observed at 1.5 hpi (Supplementary Fig. 3c). This was clarified in the results.

Figure 2c: “The number of cells in G1, S, and G2 in each subpopulation was normalized to the total number of cells in each cell-cycle phase.” Do the authors mean “normalized to the total number of cells in that population”? I suggest adding a sentence to explain the normalization performed. The authors should clearly state what “less PCNA in the nucleus” suggests, for example, “less proliferation” or possibly “cell cycle arrest”. Again, it seems like no statistical tests were performed. The cells in S phase are clearly different in the last timepoint but it is unclear if that difference is statistically significant.

- Fig. 2c was modified to show the fraction of G1-, S- and G2-cells without normalization.
- Connection of the reduced nuclear PCNA intensity to cell cycle was added in the results.
- Statistical test to compare the fraction of cells at different cell-cycle stages was added in Supplementary Fig. 4b.

Figure 2d: The figure legend does not clearly mention the statistical tests performed. Why not perform some test showing non-significant results and actually state that instead of not doing statistical tests and just stating that “there was significant overlap”? Also, why multiply by 1000?

- In Fig. 2d, violin plots were replaced with density plots to emphasize the overlap of viral marker distributions between different cell-cycle phases.
- Transcript counts per cell area were multiplied by 1,000 to have integers in the plots for visual clarity.
- In addition, R^2 of robust regression models was added to Fig. 2e to emphasize that there is no correlation between viral gene expression and cell size.

Line 146: One phenomenon hardly explains everything in most biological models. It is generally a combination of factors. This conclusion is somewhat irrelevant and is made throughout the paper.

- This is a good point, and the text was modified to emphasize that HSV-1 gene expression does not scale with the cell size as does cellular transcription. We also feel that it is important to emphasize the role of cell cycle as single-cell RNA sequencing studies have indicated that the cell cycle plays a major role in HSV-1 heterogeneity.

Supplementary figure 4 is mentioned before supplementary figure 3.

- Corrected accordingly.

When the authors perform linear regressions they choose to only report R^2 , which although the authors always correctly refer to it as “variance explained”, I believe the authors should add a sentence explaining why they based their conclusion on this single value.

- Model performance was quantified by R^2 that represents the square of correlation between the predicted and measured values, and this information was added in the results.

Minor comments:

Figure 1a,b: The size and resolution of the image and the colours chosen (green, red) make it difficult to see the UL19 marker clearly.

- Fig. 1a, b was modified accordingly. Colors were changed to magenta and cyan. In a, another example cell is shown that has high expression of both *UL29* and *UL19*. In b, only one site instead of multiple sites is shown to make cells more visible.

Page 7, line 118: Has this taken into account the time the actual infection took place?

- We appreciate that the actual infection or onset most likely takes place at different times, even when a synchronized infection protocol is used. Here, the data describe quantification of HSV-1 protein abundances at different times after the addition of the virus to cells, and it was clarified in the results that some of the heterogeneity most likely derives from different infection onset times. In order to analyse the actual infection start, one would need to do live-cell imaging but as stated above, this requires significant technological development so that multiplexed live-cell imaging could be performed.

Page 7, line 122 - 124: Were there other markers? If so, how did the others fare?

- We first used predictive power score (pps) to identify those single-cell features that correlate with uninfected/infected cell type, and only seven features gave a pps

value of at least 0.3 and these were chosen as predictors. Identification of these seven markers was clarified in the results.

Page 8, line 128: How was the cell cycle stage assessed?

- Cell cycle stage was assessed by first classifying cells into mitotic and non-mitotic using intensity and texture features of nuclear DAPI and morphology features of nucleus and cell. Then, non-mitotic cells were further classified into S-phase and non-S-phase cells using intensity and texture features of nuclear DAPI and PCNA. The non-S-phase were then classified into G1- and G2-phase cells using total nuclear intensity of DAPI. Classification is described in Methods.

Page 8, line 134: How is polymerase delta involved in the cell cycle? Stating this would be beneficial to the reader.

- This information was added in the results.

Page 8, line 137: There is still a lot of variability here that supports the idea of heterogeneous cell behavior following infection. However, it would be useful to know how many cells are involved in this measurement as the numbers are likely to change abruptly depending on them.

- Cell counts used in the analyses were added in the figure legend.

Page 9, line 157: Was the cell classification done manually? Please specify.

- All cell classifications were done using automated computer vision and computational analyses as described in methods, and this was specified in the results.

Page 9, line 158: How were the cell boundaries defined?

- Cell boundaries were defined either using total protein staining or a combination of total protein, calreticulin, and α -tubulin stainings. Cells were segmented using nuclei as seeds and the watershed transformation of the intensity images and adaptive thresholding in TissueMaps. This information is described in Methods.

Page 11, line 221: Depletion is a strong statement that is not supported by the data. "rRduction" would better represent what's been shown.

- Corrected accordingly.

Page 12, line 229: Why is there a large variability in cell numbers between infection stages?

- Classification of infected cells into different infection stages was re-done according to the reviewer #3 comments and the figure was modified accordingly (Fig. 4 in the original manuscript; Fig. 7 in the revised manuscript).

- Large variability in cell numbers between infection stages is due to the heterogenous infection progression. Infection stages were quantified at 12 hpi, and at this time point most cells are in the early and middle stages while cells at late stages are less abundant. If the same quantification was performed at a later time point, then the most abundant groups would be the late stages.

Figure 7b: Labelling the two cell types on the figure itself could be beneficial to improve clarity.

- Labels for both cell types were added in the figure.

Page 17, line 356-357: The wording is confusing

- This was clarified in the text.

Reviewer #2

In this manuscript, Pietilä and colleagues take a multiplexed imaging approach to investigate variability among cells infected by HSV-1. The work illuminates the high degree of variability during HSV-1 infection and, importantly, adds a spatial component to previously published reports by scRNAseq. I congratulate the authors on this beautiful work!

In addition to a few specific comments listed below, my main issue with the manuscript as it is currently written is the improper or not well-defined use of the term “prediction” and the improper conflation of correlation and causation. The authors extract information from cellular images and use multiple tools to correlate this information with the stage and extent of viral infection. While it is technically true that the authors use cellular features to predict the infection status of the cells, it is misleading and that this prediction does not suggest these cellular features are determinants of the infection status. Rather, these analyses show that different infection states are marked by different cellular states. Here are two examples: Lines 208 - "multivariate source of infection heterogeneity" - While the authors clearly show that infection variability correlates to variability in the multivariate cellular space, they do not show that this is the source of infection heterogeneity.

Lines 421-422 - "we identified multiple cellular factors that explain single-cell variation in human herpesvirus infection". While these cellular factors explain variability in the data, this is a correlation and not a deterministic relation.

I highly recommend the authors rephrase their statements throughout the manuscript to prevent confusion between observed correlations and their underlying causes.

- We acknowledge that our data do not reveal the direction of causality between the cellular and viral factors, and we have clarified this important point throughout the paper.

Specific comments:

1. Fig 1b, Fig 5 and Fig S4 - blurry, hard to read

- Fig. 1a, b was modified. Colors were changed to magenta and cyan. In b, only one site instead of multiple sites is shown to make cells more visible.
- Resolution of all images was improved.

2. Lines 114-119 : the authors describe that ~80% of infections originated from a single virion, and that a similar number of cells (~30%) was infected at 12 hrs and 1.5 hrs post-infection. I am confused by these numbers and would like to request the authors clarify the following points:

a. Since an MOI of 0.3 was used, how do the authors explain most cells have a single capsid in them? The ratio of genomes to PFU for HSV-1 is in the tens to hundreds, and the number of empty capsids are even higher. Wouldn't an MOI of 0.3 would translate to an average of >3 genomes per cells, and probably much more capsids?

- The titer of HSV-1 stocks was determined using immunofluorescence imaging and detection of ICP27-expressing cells. Thus, MOI was calculated using a titer that reflects successful infections. Furthermore, the virus stocks were purified through a sucrose cushion that should remove empty capsids. If the purified virus stocks contained free viral genomes, these should not be able to infect cells.
- In Supplementary Fig. 3b, only those cells that have at least one ICP5-virion were considered, and ~80% of ICP5-virion containing cells had one ICP5-virion. Supplementary Fig. 3c shows fraction of cells that contained at least one ICP5-virion (1.5 hpi time point), and most cells, ~70%, had no ICP5-virions.

b. My experience using live-imaging of HSV-1 infection suggests secondary infections are common from 6 hrs onwards. Could the authors explain why they don't see much secondary infections? Fig 1b also seems to suggest the number of infected cells increase with time.

- Both quantification of viral mRNAs (Fig. 1c and Supplementary Fig. 1a) and viral proteins (Supplementary Fig. 3f) show that the number of infected cells or cells showing HSV-1 gene expression increase with time. Our data also indicate that there are secondary infections but only at late time points (Fig. 4 and Fig. 5d). Furthermore, when using HeLa cells, we observe expression of HSV-1 late proteins, ICP5 and VP16, mainly at 9 and 12 hpi, indicating that assembly and release of new virions has only started at these time points, explaining why we don't see much secondary infections. Most likely low MOI explains why the number of secondary infections is low during the time frame used in this study.

3. From Line 235 - the section entitled "Multiplexed protein maps link cellular changes to infection heterogeneity" is unclear to me. I read this section several times, but could not understand the main points the authors were trying to make. I would suggest the authors clarify this section. However this might just be a failure on my part, so it's really up to the authors to decide if it needs clarifying.

- This section was clarified.

4. Lines 356-358 - "Notably, HSV-1 induced the formation of PBs in neither cell lines (Fig. 7c,g), although this has previously been reported in HeLa cells". This sentence is unclear. Do the authors mean infection did not induce formation of PBs in both cell lines?

- This was clarified. A previous publication indicates that HSV-1 infection induces a PB formation in HeLa cells, but we did not observe this in any of the cell lines used.

5. Lines 378-379 - the authors summarize that "HSV-1 infection progression depends on degradation of DDX6 and PBs", however this was not demonstrated in this study. While the authors have shown that during infection DDX6 and PBs are degraded, they did not show this is important for infection to progress.

- It is true that we still need to do follow-up experiments that show the dependence of HSV-1-infection progression on the observed PB loss. However, MCU data show that DDX6-enriched PB MCUs are more abundant in the infected cells that have low viral expression levels compared to infected cells with high viral expression (Fig. 4). We changed this corresponding sentence so that our data suggest such dependence.

In summary, this is an impressive and important work that further our understanding of the virus-host interactions at the single cell level, and I hope the authors find my comments useful.

- Thank you!

Kind regards,
Nir Drayman

Reviewer #3

Synopsis

Exploring and understanding heterogeneity in viral infections is key to increase our knowledge in this topic. The advent of high-throughput methods such as single-cell RNA-sequencing have greatly advanced this research, however the availability of spatial as well as protein/protein modification information is still sparse. In this study, the authors employ a multiplex immunofluorescence assay combined with a detailed, sub-cellular quantification of signal and signal patterns, in order to elucidate specific processes associated with Herpes simplex virus 1 infection in HeLa cells, such as ERK/Akt signaling and RNA polymerase II phosphorylation states. The data quality is very high, the ways the authors analyzed the data are excellent, and the biological findings of substantial interest. However, it is not clear with how many replicates the experiment have been performed, and to what extent the findings are reproducible, and also the data is often not presented adequately. Ideally, the findings should be reproduced in a second cell line such as the A549 cells employed in one experiment, or an independent experiment in HeLa cells be performed. Furthermore, clarity and level of detail of the data analysis should be improved as detailed below.

Major issues

- *It is not clear to me how many replicates (i.e. different cell culture plate wells with separate processing) were done. Basically, important observations would need to be confirmed in independent experiments.*
 - The number of different cell culture plate wells was added in the figure legends. In addition, information of independent experiments was added in the figure legends.
 - The main findings were validated in independent experiments in HeLa cells as well as in two other cell lines, A549 and BJ.

- *Figure 1/S1: the signal for UL19 is clearly weaker than UL29, presumably because the copy number is lower. This could be either an intrinsic features of UL19 mRNA, or simply because the time points are too early to get the full UL19 signal. Is the low UL19 signal in S1 due dropouts, and therefore less reliable? Or could this be a sign of e. g. an abortive replication cycle? The authors should try to explain/discuss this observation.*
 - HSV-1 *UL29* is an early gene and *UL19* a late gene, explaining why infected cells typically contain more *UL29* spots compared to *UL19* spots at the time points studied in this paper, and this was clarified in the results. Although *UL19* is less expressed compared to *UL29*, especially at 9 and 12 hpi there is also significant *UL19* expression and about half of the infected cells express *UL19*.
 - Fig. 1a and b were modified to better illustrate both *UL29* and *UL19* spots.

- *Along these lines, the raw data from the 4i experiment should be better characterized. This*

could e.g. include regression coefficients between all measured features as a supplementary table, with some (e.g. between viral proteins/RNAs, or everything that is above some threshold) highlighted in the

- We acknowledge that it would be important to describe all the features extracted from the smFISH + 4i experiment, and we have chosen to show pairwise Pearson and Spearman correlation coefficients that are above a certain threshold (Supplementary Fig. 5). However, in this study ~4,000 single-cell features were extracted from the smFISH + 4i experiment, and if a pairwise comparison of all single-cell features was shown, this would result in a correlation matrix with ~16,000,000 coefficients. Thus, it would be challenging to represent regression coefficients that cover the whole dataset. To visualize the high-dimensional cellular and infection state spaces, we have used UMAPs.

- *From Fig. 2B, it looks as if PCNA protein levels are higher in infected cells at 1.5hpi. This would indicate that cells in G2 and S phase are more prone to infection, no?*
 - Infected cells have indeed slightly increased PCNA levels compared to uninfected cells at 1.5 hpi that could indicate that S- and G2-phase cells are more susceptible to infection. Also, data in Fig. 2c support this. We also added a supplementary figure (Supplementary Fig. 4b) that shows statistical comparison of different cell-cycle phases between uninfected and infected cells, and this shows that S- and G2-phase cells are indeed statistically significantly more abundant among the infected cells. This was also added in the result text.

- *How many replicates (i.e. infected wells) make up the data in Figure 2? It would be important to see variability between independent experiments.*
 - Data in Fig. 2 are from the smFISH + 4i experiment, with four individual replicate wells that were infected with HSV-1. We then validated these data in a biologically independent experiment in HeLa, A549 and BJ cells, and this is presented in Supplementary Fig. 4. The number of wells and cell counts have now been added in figures and figure legends.

- *Figure 3 is basically very nice and interesting, however using so many different projections make it difficult to understand. On the other side, there is very little quantitative information. For example, how many cells are in the various subpopulations? Which are the defining features of the island with the arrow in Fig. 3b? Is PML body presence associated with S phase (it looks a bit when comparing the leftmost with the rightmost plot in Fig. 3g, upper row)?*
 - UMAP is a quantitative, powerful dimensionality reduction tool and reveals clusters in two-dimensional space and whether or not these clusters or patterns are associated with biological features. Thus, multiple projections were included to

illustrate the high-dimensional cellular state space of different subpopulations, and also to illustrate how the cellular landscape of the infected cells changes.

- Cell counts per subpopulation and time point were added in a supplementary figure (Supplementary Fig. 3e).
- The island indicated by the arrow in Fig. 3b (in the revised version Fig. 3a) is mainly composed of infected cells, and in Fig. 1e we show that all cellular features extracted from the smFISH + 4i experiment are required to predict an infected cell with high accuracy. Furthermore, pairwise-correlation analysis shows that none of the cellular features strongly correlated with the viral gene expression (Supplementary Fig. 5). Thus, there are most likely multiple features that in different combinations at different time points define this cell island and finding the right combination of those features among the ~3,000 single-cell features would be highly challenging. However, we also identified seven markers that were able to predict an infected cell reasonably well (Fig. 1e), and these features show such patterns on the cellular state landscape that the arrow-indicated cluster of the infected cells differ from the main cluster to some extent, and this data was added in Supplementary Fig. 6a.
- PML-body presence indeed seems to overlap with S-phase cells. However, Fig. 2c shows that cells in S-phase are the most abundant cells at 1.5 hpi and this most likely emphasizes their visibility also in the UMAP.

• *Figure 4, there is an apparent discrepancy between Fig. 4a and 4c, in that in 4a the S5P signal is constantly going down over time (i.e. shifting to the left with progressing time) whereas in 4c the “late stage infection”, which one would expect to be very prominent at 12 hpi, has again higher S5P signal. Furthermore, the cell populations are not well characterized. According to 4c, there are three populations (early/middle/late). Are these along a trajectory? What properties do these populations have, e.g. which values do the viral features in there show? Could there be more subpopulations than just these three? For example, there is quite a number of cells in 4e on the top right part of the UMAP (below the cells with lots of ICP8) with high S5P and S2P intensity, what is their status? Overall, the different cell populations need to be better characterized.*

- The discrepancy between Fig. 4a and c (in the revised version Fig. 7a and c) is due to different classification used in these figures. In Fig. 4a, infected cells contain all different infection stages that are then separated into different subgroups in Fig. 4c. Infected cells at “the latest infection stage”, which have higher S5P signal, are the smallest subgroup, even at 12 hpi, explaining why RNAP II S5P intensity seems to constantly decrease in Fig. 4a. However, there is also a significant overlap between infected and uninfected cells in Fig. 4a, and Fig. 4c reveals that this overlap is mainly due to the infected cells at early and late infection stage. One could expect that late infection stage cells are prominent at 12 hpi if a high MOI was used but here a low MOI was used, and at 12 hpi only a small subset of cells contains ICP8-positive replication compartments.

- It is true that there could be more than three subpopulations among the infected cells, and for this reason we first classified infected cells into two groups, with and without replication compartments (RCs). Then, infected cells without RCs were further clustered into four different subpopulations using mean intensity of ICP4 and ICP27. This approach resulted in five subpopulations. To characterize these subpopulations better, we now show their ICP4 and ICP27 mean intensities (Fig. 7d) and also a UMAP colored by the subpopulations (Fig. 7f).
- *Fig. 5a: Please explain the constructions of the MCU in a bit more detail so that readers do not have to go back to the original paper.*
 - The main steps of MCU constructions are now explained in the results.
- *Fig. 5: this figure contains the kind of data that I miss in Fig. 4. It is however not clear to me why cells are now classified differently compared to Fig. 4, as apparently the classification here is also along somehow along an infection trajectory. By far the best way in my opinion, in terms of clarity and traceability, would be to use the same classification of infected cells in Fig. 4 and 5, and describe the two different observations (Pol II phosphorylation, pathway activation) in the same subpopulations of cells.*
 - In Fig. 5 (in the revised version Fig. 4), a random subset of cells from the smFISH + 4i experiment was used: 5,500 cells per time point from the HSV-1-infected wells. This subsetting was necessary as the computational resources limited the number of pixels that could be analysed, and we included as many cells as possible, resulting in ~900 million pixels. However, in Fig. 4 (in the revised version Fig. 7) we wanted to analyse all cells from the smFISH + 4i experiment, and thus the same classification cannot be used in both approaches.
 - Furthermore, the classification of cells based on their MCU abundance in Fig. 5e (in the revised version Fig. 4e) was used to explore the heterogeneity in viral gene expression between infected cells, and the classification of cells in Fig. 4 (in the revised version Fig. 7) was used to explore differences in RNA Pol II phosphorylations between different subpopulations of infected cells; the main distinction is the presence of viral replication compartments. We clarified different classification approaches used in different figures by reordering figures so that the classification approach would not change from figure to figure too much.
- *It would be most interesting to see how localized Pol II signal, particularly phosphorylation, looks at viral replication centers. Can the authors add this information?*
 - Co-localization of total RNAP II, RNAP II S5P and HCFC1 in the viral replication centers is shown in Fig. 4b (in the revised version Fig. 7b) and Supplementary Fig. 11g,h. RNAP II S2P was not observed to co-localize in the replication compartments.
- *Fig. 6 – again, the infected cell subpopulations are not well described. What are e.g. the*

properties of the cells with high p-Akt? Are there category 2 cells in 6c with high p-Akt (it looks a bit like an overlap of this triangular shaped part of the UMAP below the blue island).

- In Fig. 6c (in the revised version Fig. 5c), there are indeed also subpopulation 2 cells with increased p-Akt levels. However, Fig. 6b (in the revised version Fig. 5b) shows that p-Akt levels are only slightly increased in the subpopulation 2 cells. Furthermore, this may be due to some missegmentation of cells, because in the infected cells p-Akt signal is mainly at the plasma membrane and if the cell border is slightly missegmented, the neighbors of the infected cells may get some of the signal.
- In Fig. 6 (in the revised version Fig. 5), the same cell subpopulations from HSV-1-infected wells are used as defined in Fig. 3 and also used in Fig. 4 (revised version): uninfected cells without infected neighbors, uninfected cells with infected neighbors and infected cells. Cells with high p-Akt levels are mainly among the infected cells, as shown in Fig. 6b (in the revised version Fig. 5b). At the MCU level, Fig. 4e (revised version) gives insights into the high p-Akt cells: cells in Leiden subpopulations 1 and 13. In Fig. 6 (in the revised version Fig. 5) and related supplementary figures the main aim is to show that infection activates Akt pathway in the infected cells and ERK pathway in their neighbors and why the infection results in the activation of these pathways.

Minor issues

- *Abstract, the sentences/sentence parts “we mapped high-dimensional viral and cellular state spaces throughout the infection using multiplexed imaging and quantitative single-cell measurements.” and “The multiplexed protein maps of HSV-1-induced cellular modifications uncover” should be more specific. What did you image and measure?*

- This was clarified in the abstract.

- *Introduction: Please try to rephrase the not-so-elegant first two sentences, and reconsider whether it is necessary to explicitly mention one disease (COVID-19).*

- These were rephrased.

- *Introduction: “Infections are often asymptomatic, but disease outcome can vary from mild oral herpes to severe encephalitis.” – whereas this sentence is per se not wrong, the clinical features of the key aspects primary infection/latency/reactivation should be depicted more accurately.*

- Clinical aspects were described better.

- *Fig. S2: fextures -> textures*

- Corrected.

• *Fig. S3B: how well does this distribution fit the theoretically expected distribution of entering virions?*

- In order to do model fitting to the observed distribution of entering virions, we would need to know which distribution theoretically expected distribution follows. Among the discrete distributions, does it follow for example Poisson distribution? However, theoretical considerations and modelling of entering virion distributions are out of the scope of this study.

• *Fig. 2BC, as a sanity check please show the uninfected cells from all time points next to each others with statistical tests, as there should be no difference there (except if there would be effects on neighbouring cells, but this does not seem to be the case).*

- Each time point in the smFISH + 4i experiment represents an individual 96-well plate, and some plate-to-plate variation was detected in the intensity values of different markers. This variation mostly comes from the imaging buffer that can reduce the signal intensity of some antibodies. Thus, it was essential to correct plate-to-plate variation, and this was performed by first standardizing relative to the mock cells within the same plate and then relative to the uninfected cells without the infected neighbors within the same plate. Thus, one cannot compare normalised intensities of the uninfected cells between the time points.

• *Fig. 5b and 6c, explain in the figure legends what 1/2/3 means.*

- This information was added in the figure legends.

• *Fig. 5, “Both infected cells and cells with high ICP27 expression” – what is the difference then? Are there cells with high ICP27 that are not classified as infected?*

- The purpose was to say that when infected cells are compared to uninfected cells and when cells with high ICP27 expression are compared to cells with low ICP27-expression, but our original sentence is misleading. This, this was clarified in the results.

• *Fig. 5, different column orders in 5a/e and different row orders in 5e left/right make it difficult to read. Since the clustering is not of immediate importance, it would be better to not cluster rows/columns in 5e.*

- This is a valid argument. However, we feel that clustering in Fig. 5e (in the revised version Fig. 4e) is also important to group those Leiden subpopulations that have similar MCU abundance (left) or similar viral gene expression (right) together. This will further highlight the heterogeneity between the infected cells and that Leiden subpopulations with similar viral gene expression profiles (right) have different clustering when clustered by MCU abundance (left).

- *“thus our multiplexed data reveal that HSV-1 was unable to shut down the host defense in these low-viral load cells.” infers a causality that cannot be derived from the data.*

- This was rephrased to remove causality.

- *The entry to the section “HSV-1 infection activates signaling responses also in neighboring cells” is not very elegant, please try to rephrase.*

- This was rephrased.

- *Last sentence, “...and reveals potential targets for antivirals.” – this is a bit far-fetched, and for sure does not make the paper any better.*

- This was rephrased.

REVIEWERS' COMMENTS

Reviewer #1 (Remarks to the Author):

To the Editor and Authors,

I am writing to share my review of the manuscript titled "Cellular state landscape and virus infection progression are connected" by Pietilä et al. After carefully reading the manuscript and the revisions, I am pleased to express my agreement with the author's adjustments, and I wholeheartedly recommend its acceptance and publication.

The author's revisions have significantly improved the manuscript, addressing previous concerns and enhancing the research's overall clarity. The methodology section has been refined, demonstrating the validity of the experimental design and facilitating the study's replication. The results and analysis have been presented concisely and logically, leading to clear conclusions that align with the objectives stated in the introduction.

In summary, I am impressed with the author's efforts in improving this manuscript and commend them for their hard work and dedication in addressing the previous concerns raised during the review process. The manuscript is now a valuable contribution to the field, offering new insights and expanding our understanding of viral infections and their impact on the cellular landscape. Therefore, I strongly recommend its acceptance and publication, as it will undoubtedly interest the journal's readership and positively impact the scientific community.

Thank you for considering my recommendation. I am confident that publishing this manuscript will enrich the journal's content and significantly contribute to advancing knowledge on host-pathogen interactions in the field.

Reviewer #2 (Remarks to the Author):

The authors have addressed my comments well and I have no further major comments. I have only one minor comment upon rereading of the manuscript:

In lines 151-153 the authors state: "At early times of infection, nevertheless, the multiplexed cellular information could not resolve the infection status, indicating that there was no preexisting cellular state preferred by the virus."

I think this is over interpretation. This only suggests that the cellular changes induced by HSV-1 are not apparent by the markers used here. It does not rule out "preexisting cellular state preferred by the virus".

Reviewer #3 (Remarks to the Author):

The authors have excellently improved the manuscript, and particularly the addition of the A549/BJ data is a substantial upgrade. In my opinion, the manuscript is now ready to be accepted. However, I could not check the data on Mendely, as the links provided in Supplementary Table 2 did not work. The access to this data as well as their integrity should be verified before final acceptance.

Below are some remaining minor comments.

- Considering adding herpes to the title ("Cellular state landscape and Herpes simplex virus 1 infection progression are connected")
- Figure 2, label bars with uninfected/infected as in 2b (e.g. with small indicators in the same color as in 2b), so that readers do not have to go to the figure legend in order to see what "0" and "1" means.
- Similar in Fig. S4c-e. In there, consider adding the cell line designation to the figure.

Supplementary information_Pietila et al_Nature Comminucations_FinalNCOMMS-22-46077A

Cellular state landscape and herpes simplex virus type 1 infection progression are connected
(previous title: *Cellular state landscape and virus infection progression are connected*)

Maija K. Pietilä, Jana J. Bachmann, Janne Ravantti, Lucas Pelkmans, and Cornel Fraefel

Response to referees

We appreciate the feedback from the reviewers and we have modified the manuscript accordingly. To meet the editorial team requests, the manuscript was also modified accordingly. All changes are shown in the manuscript text file with track changes.

Below please find our responses to the points raised by each reviewer.

Reviewer #1 (Remarks to the Author):

To the Editor and Authors,

I am writing to share my review of the manuscript titled "Cellular state landscape and virus infection progression are connected" by Pietilä et al. After carefully reading the manuscript and the revisions, I am pleased to express my agreement with the author's adjustments, and I wholeheartedly recommend its acceptance and publication.

The author's revisions have significantly improved the manuscript, addressing previous concerns and enhancing the research's overall clarity. The methodology section has been refined, demonstrating the validity of the experimental design and facilitating the study's replication. The results and analysis have been presented concisely and logically, leading to clear conclusions that align with the objectives stated in the introduction.

In summary, I am impressed with the author's efforts in improving this manuscript and commend them for their hard work and dedication in addressing the previous concerns raised during the review process. The manuscript is now a valuable contribution to the field, offering new insights and expanding our understanding of viral infections and their impact on the cellular landscape. Therefore, I strongly recommend its acceptance and publication, as it will undoubtedly interest the journal's readership and positively impact the scientific community.

Thank you for considering my recommendation. I am confident that publishing this manuscript will enrich the journal's content and significantly contribute to advancing knowledge on host-pathogen interactions in the field.

- Thank you!

Reviewer #2 (Remarks to the Author):

The authors have addressed my comments well and I have no further major comments. I have only one minor comment upon rereading of the manuscript:

In lines 151-153 the authors state: "At early times of infection, nevertheless, the multiplexed cellular information could not resolve the infection status, indicating that there was no preexisting cellular state preferred by the virus."

I think this is over interpretation. This only suggests that the cellular changes induced by HSV-1 are not apparent by the markers used here. It does not rule out "preexisting cellular state preferred by the virus".

- This was corrected (lines 154 and 155) to emphasize that the cellular markers used in this study do not reveal a preexisting cellular state preferred by the virus.

Reviewer #3 (Remarks to the Author):

The authors have excellently improved the manuscript, and particularly the addition of the A549/BJ data is a substantial upgrade. In my opinion, the manuscript is now ready to be accepted. However, I could not check the data on Mendely, as the links provided in Supplementary Table 2 did not work. The access to this data as well as their integrity should be verified before final acceptance.

Below are some remaining minor comments.s

- *Considering adding herpes to the title (“Cellular state landscape and Herpes simplex virus 1 infection progression are connected”*
 - *Figure 2, label bars with uninfected/infected as in 2b (e.g. with small indicators in the same color as in 2b), so that readers to not have to go to the figure legend in order to see what “0” and “1” means.*
 - *Similar in Fig. S4c-e. In there, consider adding the cell line designation to the figure.*
-
- The preview links of the Mendeley datasets are included in the author checklist.
 - We appreciate the suggestion regarding the manuscript title, and the title has been modified accordingly.
 - Figures 2 and Supplementary 4 were modified as suggested.